# SLMO transfers phosphatidylserine between the outer and inner mitochondrial membrane in *Drosophila*

**Siwen Zhao[1], Xuguang Jiang[2], Ning Li[1], Tao Wang** [ORCID][1,2,3]*

**1** College of Biological Sciences, China Agricultural University, Beijing, China, **2** National Institute of Biological Sciences, Beijing, China, **3** Tsinghua Institute of Multidisciplinary Biomedical Research, Tsinghua University, Beijing, China

* wangtao1006@nibs.ac.cn

**Data Availability Statement:** All relevant data are within the paper and its Supporting Information files.

## Abstract

Phospholipids are critical building blocks of mitochondria, and proper mitochondrial function and architecture rely on phospholipids that are primarily transported from the endoplasmic reticulum (ER). Here, we show that mitochondrial form and function rely on synthesis of phosphatidylserine (PS) in the ER through phosphatidylserine synthase (PSS), trafficking of PS from ER to mitochondria (and within mitochondria), and the conversion of PS to phosphatidylethanolamine (PE) by phosphatidylserine decarboxylase (PISD) in the inner mitochondrial membrane (IMM). Using a forward genetic screen in *Drosophila*, we found that Slowmo (SLMO) specifically transfers PS from the outer mitochondrial membrane (OMM) to the IMM within the inner boundary membrane (IBM) domain. Thus, SLMO is required for shaping mitochondrial morphology, but its putative conserved binding partner, dTRIAP, is not. Importantly, SLMO's role in maintaining mitochondrial morphology is conserved in humans via the SLMO2 protein and is independent of mitochondrial dynamics. Our results highlight the importance of a conserved PSS-SLMO-PISD pathway in maintaining the structure and function of mitochondria.

## Introduction

Mitochondria are dynamic organelles that mediate vital cellular functions such as the production of adenosine triphosphate (ATP) through oxidative phosphorylation. Thus, mitochondrial dysfunction is associated with aging and a number of human diseases such as neurodegenerative diseases, cardiovascular diseases, and liver diseases [1–3]. The mitochondrial outer and inner membranes (OMM and IMM) are comprised of various phospholipids that are critical for the proper morphology and function of mitochondria [4]. When phospholipid levels or their distribution within mitochondria are altered, mitochondrial function is disrupted, resulting in a broad spectrum of mitochondrial diseases [5].

The synthesis of phospholipids, such as phosphatidylcholine (PC), phosphatidylserine (PS), and phosphatidylinositol (PI) is restricted to the endoplasmic reticulum (ER). Phospholipids are then transported from the ER to the OMM through vesicle-independent trafficking at sites

**Funding:** This work was supported by grants from the National Natural Science Foundation of China (81870693), institution grants from the Beijing Municipal Commission of Science and Technology Commission, and the Chinese Ministry of Science and Technology awarded to T. W. The funders of the study had no role in the study design, data collection and analysis, decision to publish, or preparation of the manuscript.

**Abbreviations:** ATP, adenosine triphosphate; DP, cytidine diphosphate; CL, cardiolipin; CM, cristae membrane; CJ, cristae junction; dsRNA, double-stain RNA; ER, endoplasmic reticulum; ERG, electroretinogram; ERMCS, endoplasmic reticulum-mitochondria contact sites; FACS, fluorescence-activated cell sorting; IBM, inner boundary membrane; IBS, inner boundary space; IMM, inner mitochondrial membrane; IMS, inner membrane space; MD, molecular dynamics; MOE, Molecular Operating Environment; MRM, multiple reaction monitoring; MTBE, methyl tert-butyl ether; NAFLD, nonalcoholic fatty liver disease; OMM, outer mitochondrial membrane; OCR, oxygen consumption rate; PA, phosphatidic acid; PC, phosphatidylcholine; PE, phosphatidylethanolamine; PI, phosphatidylinositol; PISD, phosphatidylserine decarboxylase; PS, phosphatidylserine; PSS, phosphatidylserine synthase; SASA, solvent-accessible surface area; TEM, transmission electron microscopy.

called ER-mitochondria contact sites (ERMCS) [6]. An exception is phosphatidylethanolamine (PE), as most mitochondrial PE is derived from imported PS [7], which is then converted to PE in mitochondria by PISD [8,9]. Importantly, levels of these major classes of phospholipids vary widely between the ER, OMM, and IMM, suggesting mechanisms for transporting specific phospholipids between these compartments [10,11]. Recent studies have demonstrated that conserved members of the Ups family of lipid transfer proteins form complexes with Mdm35 to mediate the transport of specific phospholipids within mitochondria [12–14]. In *Saccharomyces cerevisiae*, the Ups1–Mdm35 complex facilitates the transfer of phosphatidic acid (PA) between mitochondrial membranes, while the Ups2–Mdm35 complex is involved in the transfer of PS within mitochondria. These lipid transfer activities are conserved in mammalian homologs, PRELID1-TRIAP1 and SLMO2-TRIAP1 complex [15–18]. Disturbances in mitochondrial PE synthesis disorganized cristae morphogenesis in various organisms [19–22], highlighting the importance of PE environment for maintaining mitochondrial structure. However, reduction in mitochondrial PC and cardiolipin (CL) impairs cristae morphogenesis as well [16,23]. Since PC and PE are major phospholipid in mitochondria, it is possible that overall phospholipid transport to mitochondria is fundamental to mitochondria morphology and function.

Among the major phospholipids, PE plays a key role in promoting membrane curvature of the cristae and in maintaining efficient oxidative phosphorylation [22,24,25]. Thus loss of mitochondrial PE is detrimental to health [26]. With the exception of mitochondria-generated PE, little is known about how the trafficking of specific phospholipids between the ER and mitochondria contributes to mitochondrial morphology and function. Remarkably, how the metabolism of different phospholipids is spatial and temporally coordinated to orchestrate mitochondrial biogenesis and homeostasis is poorly understood.

## Results

### Disruption of the PSS/PISD pathway affects mitochondria morphology

The structural organization and function of mitochondria rely on a continuous supply of phospholipids from the ER. We therefore asked if disrupting the synthesis of specific types of phospholipids affected mitochondrial properties. We knocked down enzymes critical for synthesizing individual phospholipids by expressing short-hairpin RNAs (*shRNA*) in muscle cells (via the combination of *MHC-gal4* and *UAS-RNAi* transgenes) and then assessed the impact on mitochondrial morphology. The major structural phospholipids are PC and PE, which are synthesized primarily from 1,2-diacyl-sn-glycerol (DAG) via the cytidine diphosphate (CDP) choline/ethanolamine pathway [27]. Disrupting the synthesis of PC and PE through *pect^RNAi^*, *pcyt1^RNAi^*, or *bbc^RNAi^* did not alter the shape or compactness of mitochondria (Fig 1A–1C). Knocking down PI synthase (PIS), which catalyzes the synthesis of PI from CDP-DAG and inositol [28], also did not alter mitochondria morphology (Fig 1A–1C). In contrast, disruption of PSS, the key enzyme in synthesizing PS in the ER, resulted in small mitochondria. Moreover, a similar reduction in mitochondria size was observed when we knocked down *pisd*, which encodes the phosphatidylserine decarboxylase that converts PS to PE in mitochondria, suggesting that PS transport is essential for maintaining mitochondrial shape (Fig 1B and 1C). To eliminate the possibility that PSS knockdown resulted in smaller mitochondria due to the accumulation of PC or PE in the ER, we conducted double knockdown experiments targeting *pss* or *pisd* with *pect* or *pcyt1*. The knockdown of *pect* or *pcyt1* in *pss^RNAi^* and *pisd^RNAi^* flies did not affect mitochondrial size and morphology (S1A Fig). Consistent with their functions, knocking down both *pss* and *pisd* reduced mitochondrial PE, and knocking down *pss* reduced mitochondrial PS, whereas loss of *pisd* resulted in the accumulation of PS in mitochondria

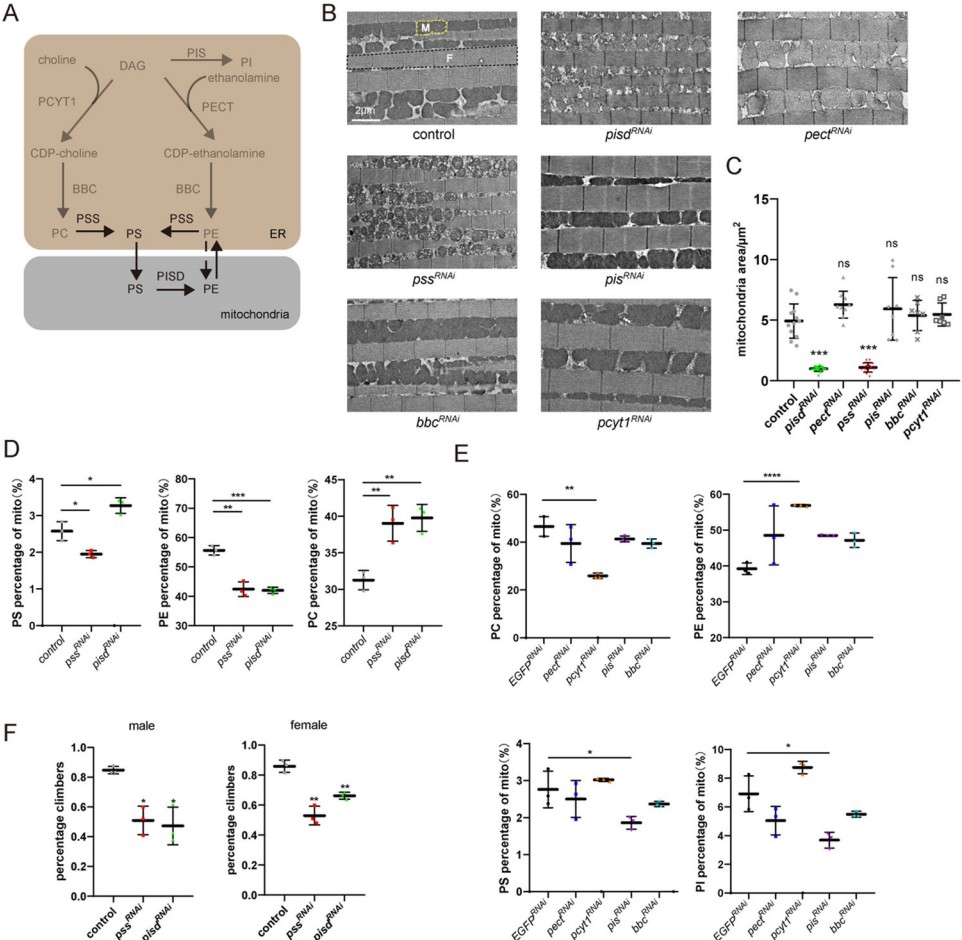

**Fig 1. The PSS/PISD pathway is involved in the structural organization of mitochondria.** (A) Schematic of phospholipid metabolism. (B) TEM sections of indirect flight muscles from control (*EGFP^RNAi*, *MHC-gal4/ UAS-EGFP^RNAi*), *pisd^RNAi* (*MHC-gal4/UAS-pisd^RNAi*), *pect^RNAi* (*MHC-gal4/UAS-pect^RNAi*), *pss^RNAi* (*MHC-gal4/UAS-pss^RNAi*), *pis^RNAi* (*MHC-gal4/UAS-pis^RNAi*), *bbc^RNAi* (*MHC-gal4/UAS-bbc^RNAi*), and *pcyt1^RNAi* (*MHC-gal4/UAS-pcyt1^RNAi*) flies. Scale bar, 2 μm. All flies were raised for 5 days before experiments. The area framed by yellow dotted lines are mitochondria labeled as "M," and the area outlined by black dotted lines are myofibrils labeled as "F." (C) Quantification of the mean area of mitochondria from (B). At least 6 samples of each genotype were analyzed. (D) Lipidomic analysis of mitochondrial phospholipid levels of control (*MHC-gal4/EGFP^RNAi*), *pss^RNAi* (*MHC-gal4/UAS-pss^RNAi*), and *pisd^RNAi* (*MHC-gal4/UAS-pisd^RNAi*) muscles. Individual phospholipid levels were calculated in molar fractions of total phospholipids. Mitochondrial were isolated from 10 dissected thoraxes per genotype. (E) Lipidomic analysis of relative mitochondrial phospholipid levels of control (*EGFP^RNAi*), *pect^RNAi*, *pcyt1^RNAi*, *pis^RNAi*, and *bbc^RNAi* muscles. Individual phospholipid levels were calculated in molar fractions of total phospholipids. Mitochondria were isolated from 10 dissected thoraxes per genotype, and 3 replicates were used. (F) Climbing ability of control, *pss^RNAi*, and *pisd^RNAi* flies. The percentage of flies that crossed the 13 cm mark within 30 s was counted. Male and female flies were calculated separately. The data underlying the graphs shown in the figure can be found in S2 Table. PISD, phosphatidylserine decarboxylase; PSS, phosphatidylserine synthase; TEM, transmission electron microscopy.

(Fig 1D). In contrast, knocking down *pect* did not affect mitochondrial PE levels, suggesting that PE from the ER may not contribute to mitochondrial biogenesis. Knocking down *pcyt1* decreased PC levels but did not affect mitochondrial size, indicating that PC is not essential for controlling mitochondrial size. Knocking down *pis* led to a significant decrease in PI, but barely affected PS and PE levels and thus mitochondrial morphology. Moreover, knocking down *bbc* did not change ratio of each phospholipid, which might due to BBC is generally required for synthesis of phospholipids on the ER (Figs 1E and S1B). Additionally, knocking

down both *pss* and *pisd* in flight muscles impaired the fly's climbing ability, indicating a loss of mitochondrial function (Fig 1F). These results agree with a previous study in both cultured cells and mice that deletion of PISD impairs mitochondrial oxidative phosphorylation (OXPHOS) and disorganized mitochondrial shape [21,22].

## SLMO is involved in maintaining mitochondria structure

Given the key role of the PSS/PISD pathway in maintaining mitochondrial structure, the transport of PS from the ER to mitochondria and within mitochondria is a major driving force for mitochondrial biogenesis. Thus, identifying the mechanisms of PS trafficking from the ER to the IMM, and/or between the OMM and IMM is important to understand PSS/PISD-dependent mitochondrial maintenance. We previously demonstrated that overexpression of PISD in *pect* mutant flies increases the synthesis of PE in mitochondria, restores overall cellular PE homeostasis, and rescues retinal degeneration and defective visual responses. This rescue is blocked by knocking down *pss* or disrupting contact between mitochondria and the ER [29,30]. Therefore, we speculated that the loss of factors that mediate the trafficking of PS could abolish the ability of PISD to compensate for *pect* loss of function. We first made 6 lines in which different *shRNA* lines targeting *pect* were driven by *GMR-gal4*, which is specific for photoreceptor cells. We found that *pect*^RNAi5^ exhibited electroretinogram (ERG) defects, specifically low amplitude responses and no off transients (S1C Fig). To rule out the possibility that the different phenotypes observed by *pect* knockdown (*pect*^RNAi^) in muscle and retina were due to differences in knockdown efficiency between these tissues, we assessed *pect* mRNA levels in the thorax and photoreceptor cells of *pect*^RNAi^ and *pect*^RNAi5^ flies, respectively. In both cases, *pect* transcript levels were reduced by approximately 40% (S1D Fig). Importantly, these ERG defects could be rescued by expressing PISD using a photoreceptor-specific *trp* (*transient receptor potential*) promoter (S1E Fig, *trp-pisd*). Disrupting ER-mitochondria contacts by *serca*^RNAi^ (*sarco/endoplasmic reticulum Ca²⁺-ATPase*) and *marf*^RNAi^ (*Mitochondrial assembly regulatory factor*) abolished the ability of *pisd* to rescue the *pect*^RNAi5^ phenotype (S1F Fig). We used this system to identify additional factors involved in PS trafficking by conducting a genome-wide RNAi screen for *RNAis* that abolished the suppressive effects of *pisd* on the *pect*^RNAi5^ phenotype (S2A Fig). From approximately 6,000 individual lines, 54 reduced ERG amplitude or transients in *pect*^RNAi5^/*pisd* flies, but had no obvious phenotype when expressed alone (S1 Table). Among these lines, *slmo*^RNAi^ efficiently reduced both ERG amplitude and transients in *pect*^RNAi5^/*pisd* flies. This phenotype was rescued by re-expressing a *slmo*^RNAi^-resistant *slmo* cDNA, as *slmo*^RNAi^ targets the 3′ UTR (Fig 2A and 2B). *GMR>pect*^RNAi5^ flies also exhibited a rapid loss of rhabdomeres and photoreceptor cells. As seen with phototransduction, this phenotype was suppressed by *trp-pisd*, and *slmo*^RNAi^ reversed this suppression (Fig 2C and 2D).

SLMO in *Drosophila* has been reported to be essential for peristaltic movement and germline proliferation, but its exact function remains unclear. We observed that SLMO is homologous to yeast proteins Ups1/2 and the human PRELI family proteins (PRELID1, SLMO1, and SLMO2). Ups1/PRELID1 was shown to be crucial for maintaining CL levels by mediating PA transport, whereas Ups2/SLMO2 is capable of transferring PS [15,16]. As fly SLMO is more similar to Ups2, we speculated that SLMO may be involved in PS transport and the structural organization of mitochondria. To evaluate the function of SLMO in vivo, we first generated an *slmo* null mutant (*slmo*^1^) using CRISPR/Cas9 genome editing (S2B Fig). *slmo*^1^ homozygous mutants failed to develop and exhibited a growth arrest at the L1 larval stage (S2C Fig). Using mitotic recombination (*ey-flp/hid* system) [31], we generated homozygous *slmo*^1^ mutant eyes (in an otherwise heterozygous body) and found that the *slmo*^1^ mutation caused a cell-

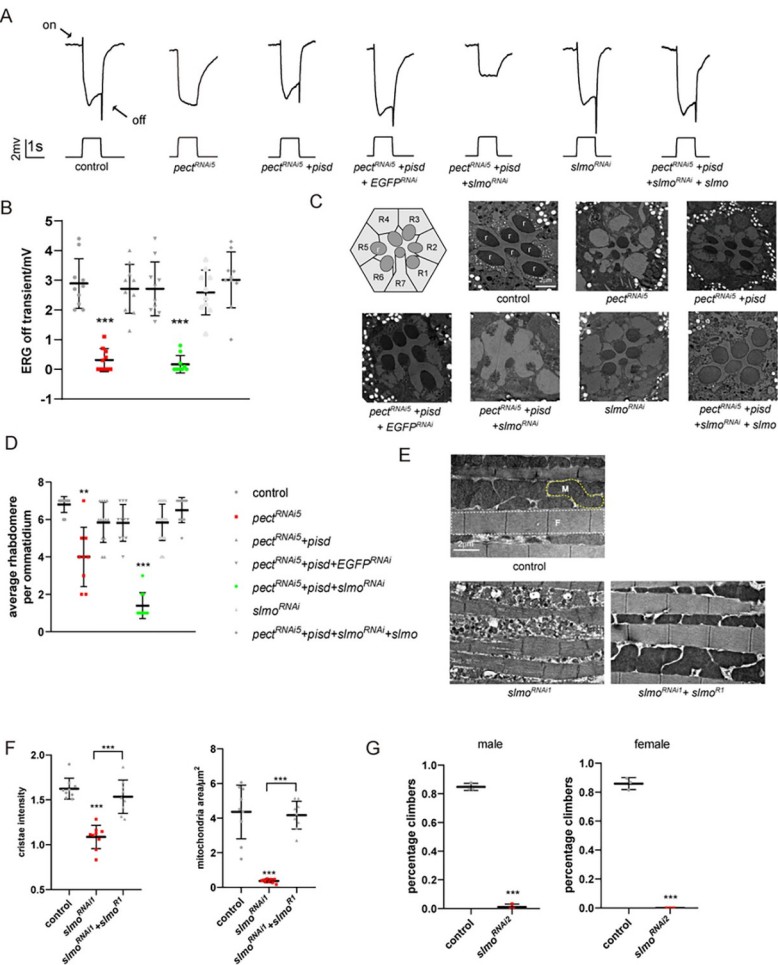

**Fig 2. SLMO is required for PSS/PISD to maintain mitochondrial structure.** (A, B) *slmo^RNAi* impaired the ERG responses of *pect^RNAi5*/*pisd* flies. Five-day-old control (*EGFP^RNAi*, *GMR-Gal4/UAS-EGFP^RNAi*), *pect^RNAi5* (*GMR-Gal4 /UAS-pect^RNAi5*), *pect^RNAi5*+*pisd* (*GMR-Gal4 trp-pisd/UAS-pect^RNAi5*), *pect^RNAi5*+*pisd*+*EGFP^RNAi* (*GMR-Gal4 trp-pisd/ +;UAS-pect^RNAi5/UAS-EGFP^RNAi*), *pect^RNAi5*+*pisd*+*slmo^RNAi* (*GMR-Gal4 trp-pisd/+;UAS-pect^RNAi5/UAS-slmo^RNAi*), *slmo^RNAi* (*GMR-Gal4/UAS-slmo^RNAi*), and *pect^RNAi5*+*pisd*+*slmo^RNAi*+*slmo* (*GMR-Gal4 trp-pisd/UAS-slmo;UAS-pect^RNAi5/UAS-slmo^RNAi*) flies were dark adapted for 2 min and then exposed to a 5-s pulse of orange light. On- and off-transients are indicated. (B) The off-transients, which reflect visual transmission, were quantified. At least 10 flies were used, and differences were assessed using the unpaired *t* test. (C) TEM sections were obtained from the retina of 10-day-old flies of the same genotype as (A). Cartoon picture of cross-sections through the distal regions of the ommatidia illustrates R1-R7 photoreceptor cells in a single ommatidium. r, rhabdomeres. Scale bar, 2 μm. (D) Quantification of rhabdomeres per ommatidium of from TEM sections shown in (C). At least 10 ommatidia from each section of 3 different eyes were quantified for each genotype. (E) Muscles sections of control (*MHC-gal4/UAS-GFP*), *slmo^RNAi1* (*MHC-gal4/UAS-slmo^RNAi1*), and *slmo^RNAi1*+*slmo^R1* (*MHC-gal4/+;UAS-slmo^RNAi1/UAS-slmo^R1*) flies. M, Mitochondria; F, myofibril. Scale bar, 2 μm. (F) Quantification of cristae density and mitochondrial area from (E). At least 6 samples of each genotype were assessed. Data are presented as mean ± SD, **$p < 0.01$, ***$p < 0.001$ (Student's unpaired *t* test). (G) The climbing ability of *MHC>EGFP^RNAi* (control) and *MHC>slmo^RNAi2* flies depicted as the percentage of flies that crossed the 13 cm mark within 30 s. Male and female flies were calculated separately. The data underlying the graphs shown in the figure can be found in S1 Data. ERG, electroretinogram; PISD, phosphatidylserine decarboxylase; PSS, phosphatidylserine synthase; TEM, transmission electron microscopy.

autonomous growth arrest (S2D Fig). To assess phenotypes of stronger *slmo* alleles, we generated 2 additional *slmo^RNAi* lines using the *pNP* system [32,33]. Unlike *slmo^RNAi*, which reduced *slmo* mRNA levels to 64%, *slmo^RNAi1* or *slmo^RNAi2* reduced *slmo* mRNA levels by 80% and 60%, respectively (S2 Table). Expressing *slmo^RNAi1* or *slmo^RNAi2* in photoreceptor cells

($GMR{>}slmo^{RNAi1}$ or $GMR{>}slmo^{RNAi2}$) diminished ERG responses and eliminated all contact between photoreceptor cells and rhabdomeres. Moreover, the ERG and degeneration phenotypes associated with $GMR{>}slmo^{RNAi1}$ were rescued by expressing an $slmo^{RNAi1}$-resistant $slmo$ cDNA (S2E–S2G Fig). We further examined the role of SLMO in mitochondrial morphology. As seen with $pss$ and $pisd$, knocking down $slmo$ via $slmo^{RNAi1}$ in muscle resulted in smaller mitochondria with damaged cristae (Fig 2E and 2F), and other lines of $slmo$ knocking down also affected mitochondrial morphology to varying degrees (S2H Fig). Expressing an $slmo^{R-NAi1}$-resistant $slmo$ cDNA in this context restored mitochondrial morphology (Fig 2E and 2F). Because the $MHC{>}slmo^{RNAi1}$ line failed to development for more than 5 days, which is unavailable for locomotor activity assay, we used $MHC{>}slmo^{RNAi2}$, which also resulted in a climbing phenotype (Fig 2G).

## SLMO functions in the PSS/PISD pathway

To further understand the in vivo function of SLMO in the PSS/PISD pathway, we performed an epistatic analysis. Although lacking $pss$ in the retina did not cause a noticeable degeneration, knocking down $pss$ resulted in reduced mitochondrial size in muscle. Overexpression of SLMO in this context rescued mitochondrial size of $pss^{RNAi}$ cells, but did not affect mitochondrial morphology in the wild-type background (Fig 3A and 3B). Consistent with the mitochondrial morphology, PS and PE levels of PSS-deficient mitochondria were also elevated by either re-expression of PSS or overexpression of SLMO (S3A Fig). In contrast, muscles expressing $slmo^{RNAi}$ exhibited a mitochondrial phenotype, which was not rescued by PSS overexpression (Fig 3C). Since overexpression of PISD in muscle caused severe mitochondrial defects, we performed the PISD epistatic analysis in the retina. Expressing a relatively weak $slmo$ RNAi line $slmo^{RNAi5}$, which reduced $slmo$ mRNA level to about 55% (S2 Table), in photoreceptors resulted in reduced ERG amplitude and transients. Overexpression of PISD in this context via the $trp$ promoter rescued these phenotypes (Fig 3D and 3E). Moreover, overexpression of PISD prevented rhabdomere loss and retinal degeneration seen in aged $slmo^{RNAi5}$-expressing flies (Fig 3F and 3G). Consistent with this phenotype, knocking down of $slmo$ in the retina also resulted in a significant decrease in PE; this reduction in PE content could also be restored by PISD overexpression (S3B Fig). In contrast, overexpression SLMO failed to prevent the mitochondrial fragmentation of the $MHC{>}pisd^{RNAi}$ cells (S3C Fig). These epistatic analyses provide strong in vivo evidence that SLMO functions downstream of PSS and upstream of PISD.

## SLMO localizes to the inner boundary space (IBS) of mitochondria

Since PSS and PISD function on the ER and IMM, respectively, we next examined the subcellular localization of SLMO, asking whether SLMO contributes to ER/mitochondria or OMM/IMM communication. Since we failed to generate a high-affinity SLMO antibody, we expressed in vivo a version of SLMO with a C-terminal GFP tag using the $ubiquitin$ ($ubi$) promoter. Importantly, $ubi$-$slmo^{R1}$-$GFP$, which is resistant to $slmo^{RNAi1}$, rescued the lethality of $slmo^1$ mutants and the ERG phenotypes of $slmo^{RNAi1}$ flies (S4A and S4B Fig). Thus, the GFP tag did not affect the SLMO function. We examined multiple tissues in $ubi$-$slmo^{R1}$-$GFP$ flies, including muscle and testis, and found that SLMO-GFP localized to mitochondria (Fig 4A). Mitochondria undergo 5 key steps during sperm formation: aggregation, fusion, membrane wrapping to form nebenkerns, unfurling, and elongation. During the onion stage of early testis development, SLMO and TOM20 co-localize on the nebenkerns, which are enlarged mitochondrial derivatives [34]. Of note, TOM20 diminished in mature sperm, while SLMO was still expressed (Fig 4A). Given the high demand for mitochondrial membrane phospholipids in each stage of spermatogenesis, it is reasonable that SLMO remains present in the

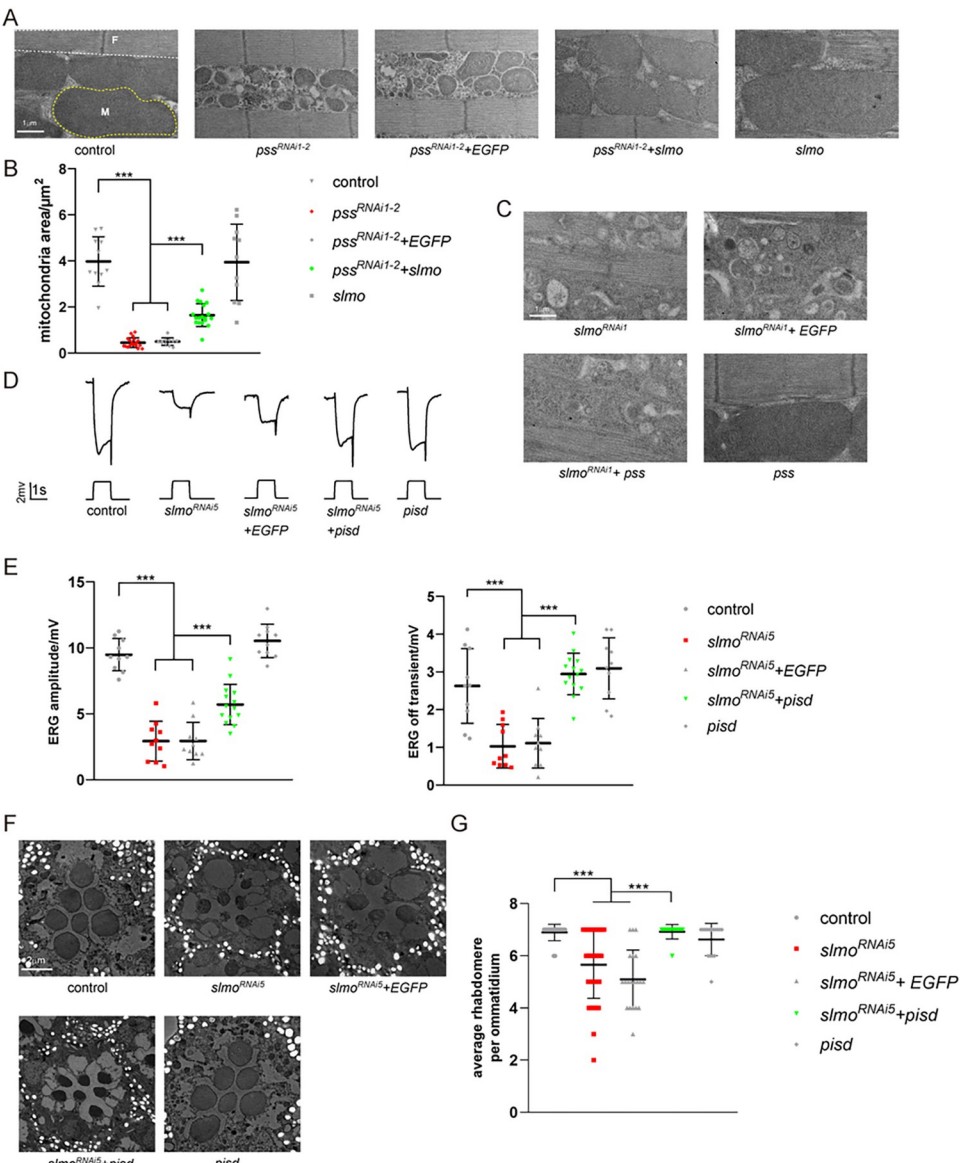

**Fig 3. SLMO functions downstream of PSS and upstream of PISD.** (A) TEM sections from muscles of control (MHC-gal4/UAS-GFP), pss$^{RNAi1-2}$ (MHC-gal4/UAS-pss$^{RNAi1-2}$), pss$^{RNAi1-2}$+EGFP (MHC-gal4/+;UAS-pss$^{RNAi1-2}$/UAS-EGFP), pss$^{RNAi1-2}$+slmo (MHC-gal4/+;UAS-pss$^{RNAi1-2}$/UAS-slmo), and slmo (MHC-gal4/UAS-slmo) flies. Scale bar, 1 μm. (B) Quantification of the mean area of mitochondria from (A). At least 6 samples of each genotype were used for quantification. (C) Overexpression of PSS failed to rescue slmo$^{RNAi}$ in terms of mitochondria morphology. Muscles sections of slmo$^{RNAi1}$ (MHC-gal4/UAS-slmo$^{RNAi1}$), slmo$^{RNAi1}$+EGFP (MHC-gal4/+;UAS-slmo$^{RNAi1}$/UAS-EGFP), slmo$^{RNAi1}$+pss (MHC-gal4/+;UAS-slmo$^{RNAi1}$/UAS-pss), and pss (MHC-gal4/UAS-pss) flies. Scale bar, 1 μm. (D) Expression of PISD rescued the ERG defects caused by slmo$^{RNAi}$. Five-day-old control (GMR-gal4/UAS-EGFP$^{RNAi}$), slmo$^{RNAi5}$ (GMR-gal4/UAS-slmo$^{RNAi5}$), slmo$^{RNAi5}$+EGFP (GMR-Gal4/+;UAS-slmo$^{RNAi5}$/UAS-EGFP), slmo$^{RNAi5}$+pisd (GMR-Gal4/+;UAS-slmo$^{RNAi5}$/trp-pisd), and pisd (trp-pisd) flies were dark adapted for 2 min and then exposed to a 5-s pulse of orange light. (E) The amplitudes of ERG response and off-transients were quantified. At least 10 flies were used, and significant differences were determined using the unpaired *t* test. (F) TEM retina sections were obtained from 10-day-old flies of the same genotype as (D). Scale bar, 2 μm. (G) Quantification of rhabdomeres per ommatidium in genotypes indicated. At least 10 ommatidia from each section of 3 different eyes were quantified for each genotype. The data underlying the graphs shown in the figure can be found in S1 Data. ERG, electroretinogram; PISD, phosphatidylserine decarboxylase; PSS, phosphatidylserine synthase; TEM, transmission electron microscopy.

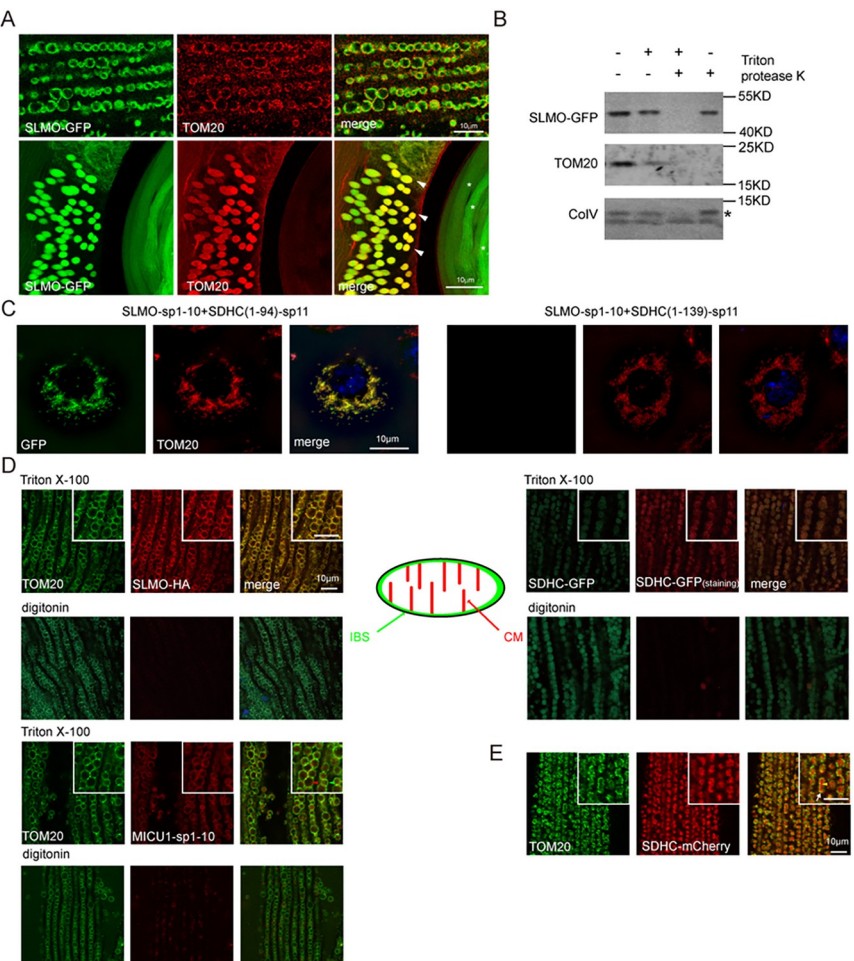

**Fig 4. SLMO localizes to the inner boundary space of mitochondria.** (A) SLMO localizes to mitochondria. Muscle and testis tissues of *ubi-slmo-GFP* flies were dissected and stained with GFP (green) and TOM20 (red). The Arrow head indicates aggregated mitochondria within the Nebenkern at the onion stage. Asterisks are used to mark elongated mitochondria in mature spermatozoa. Scale bar, 10 μm. (B) Cell lysates from S2 cells transfected with SLMO-GFP were analyzed using the protease K protection assay. Without Triton X-100, SLMO and CoIV were resistant to protease K treatment, whereas TOM20 was sensitive. (C) S2 cells expressing SLMO-sp1-10 and SDHC(1–94)-sp11 or SDHC(1–139)-sp11 were imaged using confocal fluorescent microscopy. TOM20 (red) was stained to visualize mitochondria. The scale bar represents 10 μm. (D) Super-resolution SIM microscopy localizes SLMO to the IBM. Muscle tissues from *slmo-HA* flies, *ubi-MICU1-sp1-10* (could be stained by antibody anti-GFP), and *ubi-SDHC(1–94)-GFP* flies were penetrated by Triton X-100 or 5 μg/ml digitonin, followed by staining for SLMO-HA and MICU1-sp1-10 (left, red), TOM20 (left, green), and GFP (right, red). Scale bar, 10 μm. The OMM (green arrowhead) and IBS (red) regions of IMS of mitochondria are illustrated. (E) Staining of mitochondria from muscles of *ubi-SDHC(1–94)-mCherry* by TOM20 (green) and mCherry (red) antibodies. The OMM (white arrow) and CM (yellow arrow head) of mitochondria are illustrated. The tissues were penetrated by Triton X-100. Scale bar, 10 μm. CM, cristae membrane; IBS, inner boundary space; IMS, inner membrane space.

mitochondria of mature sperm, reflecting key SLMO's role in maintaining phospholipid levels during mitochondrial elongation [35]. Therefore, there are different quality controls for TOM20 and SLMO proteins in *Drosophila* spermatogenesis. We also expressed SLMO-GFP in S2 cells (S4C Fig) where it colocalized with the mitochondrial marker, TOM20, but not with the ER marker, calnexin (CNX). We next asked whether SLMO was inside mitochondria using the protease K protection assay in S2 cells. As seen with the IMM protein (CoIV), IMS protein (SDHC(1–94)-mCherry), and matrix protein (Cox8-Sod2-GFP, mito-GFP), SLMO-GFP was

degraded only when detergent was added, suggesting that SLMO localized within mitochondria (Figs 4B and S4D). We next used a split-GFP system (in which proteins are tagged with spGFP1-10 or spGFP11) to ask if SLMO localized to the inner membrane space (IMS) or matrix [36,37]. To verify the system, we fused spGFP11 to SDHC(1–94) (succinate dehydrogenase complex subunit C residues 1–94, which localizes to the IMS [38]) and SDHC(1–139) (which localizes to the matrix [38]). These were co-expressed with spGFP1-10-tagged MICU1, which localizes to the IMS, or spGFP1-10-tagged Cox8, which localizes to the matrix. GFP fluorescence was detected when S2 cells expressed SDHC(1–94)-sp11/MICU1-sp1-10 or SDHC(1–139)-sp11/Cox8-sp1-10 [38] (S4E and S4F Fig). Similar to MICU1, the GFP signal was only detected when SLMO-sp1-10 was co-transfected with SDHC(1–94)-sp11, suggesting that SLMO localizes exclusively to the IMS (Fig 4C). To determine the localization of endogenous SLMO, we inserted an HA-tag into the c-terminus of the *slmo* genomic locus (S4G Fig). Importantly, homozygous *slmo-HA* flies were indistinguishable from wild type, indicating that the HA tag did not affect SLMO function. The SLMO-HA signal colocalized with TOM20 in muscles treated with Triton-X100. Under low concentration of digitonin, the OMM protein TOM20 could be labeled, whereas the IMM protein SDHC remained unlabeled. Higher concentration of digitonin (0.1 mg/ml) penetrated the OMM, resulting in labeling of IMM proteins (S4H Fig). Notably, SLMO-HA and MICU1-sp1-10 were detected only when using Triton, but not 5 μg/ml digitonin, which is insufficient to penetrate the mitochondrial membrane (Fig 4D). The mitochondria IMS is divided into 2 structurally and functionally distinct compartments, namely the cristae membrane (CM) and the inner boundary space (IBS), which lies between the OMM and IMM. Mitochondrial Ca$^{2+}$ uniporter component MICU1 exclusively localizes at the IMS, while OXPHOS complex is enriched in CM [39]. Using super-resolution microscopy, we found a circular pattern of SLMO-HA and MICU1-sp1-10 that co-localized with TOM20. By contrast, the CM protein SDHC (SDHC(1–94)-GFP or mCherry), which is enriched in the CM region, was surrounded by TOM20 (Fig 4D and 4E). These results indicated that SLMO localized exclusively to the IBS, consistent with a putative function for SLMO in mediating phospholipid exchange between the inner and outer mitochondrial membranes.

## SLMO transports PS in vitro and in vivo

Next, to assess the lipid transport activity of SLMO we created and purified a version of SLMO with an N-terminal MBP tag from *E. coli* (S5A Fig). The MBP-SLMO fusion protein, but not MBP alone bound specifically to PS on the Lipid Strip (S5B Fig). We confirmed that SLMO binds PS using a liposome floatation assay. After ultracentrifugation, SLMO was found within the PS-containing liposome in the upper components, whereas MBP was in the lower components (S5C Fig). We next directly examined the lipid transport ability and specificity of SLMO using a lipid transfer assay. Consistent with its binding specificity, SLMO efficiently transported PS but not PE between liposomes in vitro (Fig 5A). Using MOE (Molecular Operating Environment) to dock PS to SLMO, and MD (molecular dynamics) to simulate the binding of PS to SLMO, we found that Thr93 may form a hydrophobic interaction and Asn150 may form a hydrogen bond with PS, thus both sites are predicted to be important for transporting PS (Fig 5B). Both Asn150 and Thr93 (Ser in SLMO2 and Ups2) residues are conserved in human SLMO2 and yeast Ups2 and were indicated as key phospholipid-binding sites (S5D Fig) [17]. Based on the structural analysis, we mutated both Thr93 and Asn150 to Ala, and both SLMO$^{T93A}$ and SLMO$^{N150A}$ proteins reduced PS binding ability and failed to transport PS in vitro (Figs 5C and S5E). Using these 2 mutant proteins, we asked whether the PS transporting activity of SLMO is required to maintain mitochondrial morphology and activity. To avoid overexpression effects, we expressed different forms of SLMO in vivo using the weak and

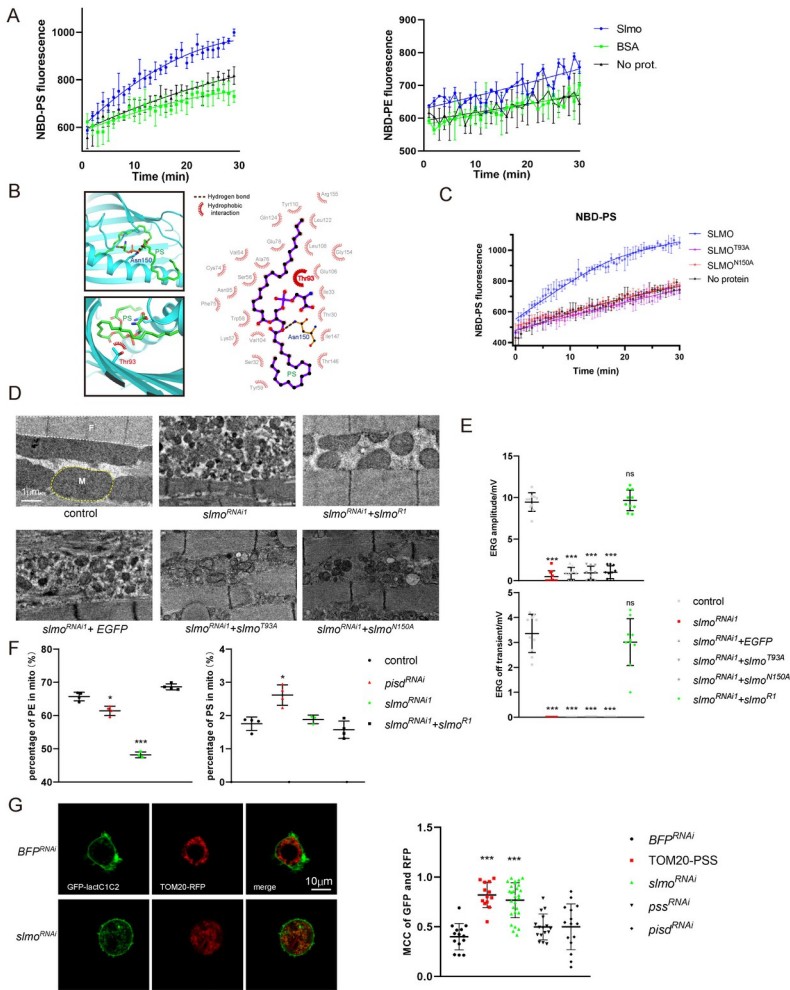

**Fig 5. SLMO transports PS from the outer to the inner mitochondrial membrane.** (A) SLMO transfer PS between liposomes. A schematic diagram of the lipid transfer assay is shown on the left, in which donor liposomes with NBD-PS or NBD-PE, and acceptor liposomes were incubated with MBP-SLMO. Time courses of normalized fluorescence signals from liposome mixtures containing NBD-PS or NBD-PE in the donor liposomes with SLMO or BSA are plotted. (B) Design of mutant SLMO in PS harboring. From MD calculation, Thr93 and Asn150 interact with PS, thus mutating both Thr93 and Asn150 to Ala may prevent PS binding in the pocket. (C) SLMO$^{T93A}$ and SLMO$^{N150A}$ abolish the PS transporting ability. Time courses of normalized fluorescence signals from liposome mixtures containing NBD-PS in the donor liposomes with SLMO, SLMO$^{T93A}$, or SLMO$^{N150A}$ are shown. (D, E) SLMO$^{T93A}$ and SLMO$^{N150A}$ are not able to replace SLMO in promoting mitochondrial organization and cellular function. (D) Muscles sections of control (*MHC-gal4/UAS-GFP$^{RNAi}$*), *slmo$^{RNAi}$* (*MHC-gal4/UAS-slmo$^{RNAi1}$*), *slmo$^{RNAi1}$+EGFP* (*MHC-gal4/+;UAS-slmo$^{RNAi1}$/UAS-GFP*), *slmo$^{RNAi1}$+slmo$^{R1}$* (*MHC-gal4/+;UAS-slmo$^{RNAi1}$/DA-slmo$^{R1}$*), *slmo$^{RNAi1}$+slmo$^{T93A}$* (*MHC-gal4/+;UAS-slmo$^{RNAi1}$/DA-slmo$^{T93A}$*), and *slmo$^{RNAi1}$+slmo$^{N150A}$* (*MHC-gal4/+; UAS-slmo$^{RNAi1}$/DA-slmo$^{N150A}$*) flies. M, Mitochondria; F, myofibrils. Scale bar, 1 μm. (E) Quantification of ERG amplitudes and off-transients of control (*GMR-gal4/UAS-GFP$^{RNAi}$*), *slmo$^{RNAi}$* (*GMR-gal4/UAS-slmo$^{RNAi1}$*), *slmo$^{RNAi1}$+EGFP* (*GMR-gal4/+;UAS-slmo$^{RNAi1}$/UAS-GFP*), *slmo$^{RNAi1}$+slmo$^{T93A}$* (*GMR-gal4/+;UAS-slmo$^{RNAi1}$/UAS-slmo$^{T93A}$*), *slmo$^{RNAi1}$+slmo$^{N150A}$* (*GMR-gal4/+;UAS-slmo$^{RNAi1}$/DA-slmo$^{N150A}$*), and *slmo$^{RNAi1}$+slmo$^{R1}$* (*GMR-gal4/+; UAS-slmo$^{RNAi1}$/DA-slmo $^{R1}$*) flies. ERG from at least 10 flies was used, and significant differences were determined using the unpaired *t* test. *DA-slmo $^{R1}$*, *DA-slmo$^{T93A}$*, and *DA-slmo$^{N150A}$* were designed as *slmo$^{RNAi1}$* resistant. (F) Lipidomic analysis of mitochondrial PE and PS levels of control (*MHC-gal4/GFP$^{RNAi}$*), *pisd$^{RNAi}$* (*MHC-gal4/UAS-pisd$^{RNAi}$*), *slmo$^{RNAi1}$* (*MHC-gal4/UAS-slmo$^{RNAi1}$*), and *slmo$^{RNAi}$+slmo$^{R1}$* (*MHC-gal4/+;UAS-slmo$^{RNAi}$/UAS-slmo$^{R1}$*) muscles. Mitochondria were isolated from 10 dissected thoraxes per assay, and 4 replicates were quantified. (G) Knocking down *slmo* in S2 cells increased PS levels in the OMM. Live images of cells transfected with Tom20-PSS (red), GFP-LactC1C2 (green), and dsRNA of *BFP* (control) or *slmo* (*slmo$^{RNAi}$*). Scale bar, 10 μm. Manders' Colocalization Coefficients (MCC) of GFP-LactC1C2 and Tom20-RFP of S2 cells transfected with *BFP$^{RNAi}$*, TOM20-PSS, *slmo$^{RNAi}$ pss$^{RNAi}$*, or *pisd$^{RNAi}$* were quantified by ImageJ. The data underlying the graphs shown in the figure can be found in S1 Data. ERG, electroretinogram; MD, molecular dynamics; OMM, outer mitochondrial membrane; PE, phosphatidylethanolamine; PS, phosphatidylserine.

ubiquitous *DA* (daughterless) promoter. Wild-type SLMO, SLMO[T93A], and SLMO[N150A] were all stably expressed and localized to mitochondria (S5F and S5G Fig). Importantly, wild-type SLMO (*DA-slmo*) rescued the lethality of the *slmo[1]* mutant, whereas SLMO[T93A] and SLMO[N150A] did not (S5H Fig). SLMO[T93A] and SLMO[N150A] also failed to rescue the mitochondrial morphological phenotypes and defective ERG responses caused by *slmo[RNAi1]* (SLMO[T93A] and SLMO[N150A] are *slmo[RNAi1]* resistant). These phenotypes were fully rescued by wild-type SLMO (Figs 5D, 5E, and S5I).

We next assessed the role of SLMO in regulating mitochondrial phospholipid composition in vivo. As seen with knocking down *pss* and *pisd*, knocking down *slmo* reduced mitochondrial PE. This was rescued by re-introducing SLMO (Fig 5F). However, knocking-down *slmo* did not increase total mitochondrial PS. If SLMO were to mediate the transfer of PS between the OMM and IMM, PS should accumulate in the OMM in response to *slmo[RNAi]*. To assess PS levels in the OMM, we generated an intracellular PS reporter by fusing the PS binding C1C2 domain of mouse cadherin (LactC1C2) to GFP and transfected this construct into S2 cells [40,41]. The LactC1C2 domain has been shown to bind to the head group of PS, which tends to form a phospholipid bilayer in the aqueous phase that hides the acetyl chain inside and the hydrophilic head is exposed [42]. Therefore, the cytosolic LactC1C2 reporter can detect PS on the outer leaflet endomembrane system as long as the inner leaflet of the plasma membrane. In control cells, GFP-LactC1C2 was distributed diffusely throughout the cytosol and vesicles, with enrichment on the plasma membrane but completely absent from mitochondria. Elevating the level of PS in the OMM by expressing a mitochondria-anchored PSS led to the recruitment of GFP-LactC1C2 to mitochondria. Importantly, SLMO knockdown resulted in the translocation of GFP-LactC1C2 to mitochondria, whereas *pisd[RNAi]* failed to induce this translocation despite elevating total mitochondrial PS (Figs 5G and S6A). Co-transfected IMM marker Cox8 exhibited a solid round pattern within the circle of LactC1C2 and TOM20, which means PS accumulated on OMM was detected (S6B Fig). These data indicate that PS trafficking from the OMM to the IMM is prevented by the loss of SLMO, leading to PS accumulation on the outer leaflet of the OMM. Collectively, we conclude that SLMO localizes to the mitochondrial inner boundary space, functioning to transport PS from the OMM to the IMM. This phospholipid flux is essential for maintaining mitochondrial morphology and function. To exclude the possibility that *slmo[RNAi]*-mediated changes in mitochondria morphology resulted from altered fusion and fission of mitochondria, we asked whether induction of mitofission or mitofusion (by overexpressing DRP1 or MFN (MARF), respectively) affected these phenotypes. MFN or DRP1 overexpression did not affect *slmo[RNAi2]*-induced reductions in mitochondrial cristae (S6C Fig). Further, Opa1 regulates cristae formation and inner membrane fusion [43,44]. Thus, we asked whether the alterations in mitochondria morphology induced by *slmo[RNAi]* were associated with OPA1. However, overexpression of OPA1 did not affect mitochondria morphology in *slmo[RNAi]* flies (S6D Fig). Therefore, it appears that SLMO influences mitochondria morphology independently of mitofission or mitofusion.

## The function of SLMO is independent of dTRIAP1 and dTRIAP2

In both yeast and mammals, the PRELI family of proteins relies on their binding partner, Mdm35/TRIAP1 [13,17,18]. In *Drosophila*, CG30108 and CG30109 are 2 homologs of Mdm35/TRIAP1. They reside next to one another in the *Drosophila* genome and may serve redundant roles due to genetic duplication during evolution. We therefore named the *CG30108* and *CG30109* gene products dTRIAP1 (d̲rosophila T̲p53 R̲egulated I̲nhibitor of A̲poptosis 1) and dTRIAP2, respectively. To determine whether dTRIAP1/2 is involved in SLMO function, we generated a *dtriap1/2[KO]* allele, in which the *CG30108* and *CG30109* loci are both deleted (Fig 6A). Unlike *slmo* mutants, animals lacking *dtriap1* and *dtriap2* were viable, with

normal mitochondrial morphology and ERG responses (Fig 6B and 6C). Neither PS nor PE were affected in the *dtriap1* and *dtriap2* double knock-out line (Fig 6D). Overexpression of dTRIAP1 or dTRIAP2 did not influence phenotypes associated with *slmo^RNAi1^* (S7A Fig), and co-expression of dTRIAP1 or dTRIAP2 with SLMO did not result in a phenotype (Figs 6E and S7B). The loss of *dtriap1*/2 also failed to modify the severity of mitochondrial phenotypes of the *slmo^RNAi^* muscle and did not reduce protein levels of SLMO (Figs 6F and S7C–S7E). We next assessed the subcellular localization of dTRIAP1 and dTRIAP2 in S2 cells and found that both dTRIAP1 and dTRIAP2 localized to mitochondria. To rule out potential impacts of the tag on dTRIAP1/2 expression and function in S2 cells, we produced an antibody targeting dTRIAP1 and observed that the C-terminal tag had no discernible effect on dTRIAP1 localization (Fig 6G). However, they localized to the OMM facing the cytosol, as split-GFP signals were only detected when dTRIAP1-sp1-10 or dTRIAP2-sp1-10 were transfected with TOM20-sp11 but not with SDHC(1–139)-sp11 or SDHC(1–94)-sp11 (Fig 6H). Additionally, we employed a protease K protection assay, revealing that dTRIAP1 resides exclusively on the OMM, as it was completely digested by protease K in the absence of Triton (Fig 6I). Therefore, fly SLMO does not require dTRIAP1 or dTRIAP2 to transport PS between the OMM and IMM.

## The function of SLMO is conserved in mammals

As the PRELI family of proteins plays a conserved role in lipid transport [17,18], we asked if SLMO's ability to transport PS between the OMM and IMM is conserved in mammals. In humans, there are 3 members of the PRELI family: PRELID1, SLMO1, and SLMO2. We first examined if these proteins could functionally substitute for SLMO when expressed in flies. We generated transgenes to express PRELID1, SLMO1, or SLMO2 in *slmo^RNAi^* photoreceptor cells. Importantly, SLMO2 fully restored the ERG amplitude and transients in *slmo^RNAi1^* flies, whereas PRELID1 and SLMO1 did not (Figs 7A and S8A). This is consistent with the result previously, which found that SLMO2 could rescue PE levels in Δups2 cells in yeast [15]. To confirm that SLMO2 is the bona fide mammalian equivalent of SLMO, we explored the localization of SLMO2. SLMO2-GFP and PRELID1-GFP localized to mitochondria when expression in Hela cells, whereas SLMO1-GFP was ubiquitously distributed throughout the cytosol (Figs 7B and S8B). When penetrating the plasma membrane with digitonin, PRELID1 co-localized with mitochondrial signals, whereas no SLMO2 signals were detected. This indicated that SLMO2 is inside mitochondria and that PRELID1 is on the OMM (Fig 7C). Using the split-GFP system, GFP signals were detected only when Hela cells were co-transfected with SLMO2-spGFP1-10 and SDHC(1–94)-spGFP11 but not with SDHC(1–139)-spGFP11, demonstrating that SLMO2 localized exclusively to the IMS (Figs 7D and S8C). Finally, we knocked down SLMO2 in Hela cell using 2 different *shRNAs* (*Slmo2^RNAi1^* and *Slmo2^RNAi2^*) and found that similar to *slmo* mutant flies, loss of SLMO2 resulted in dramatically smaller mitochondria with disrupted cristae structures. In addition, mitochondrial activity was reduced, as measured by oxygen consumption rate (OCR). These phenotypes were rescued by re-introducing SLMO2 (Figs 7E–7G, S8D, and S8E). We further confirmed that SLMO2 is not involved in mitochondrial fusion/fission by calculating the ratio of mitochondrial fusion and fission in *Slmo2* mutant cells. As a positive control, MFN1 knock-out cells exhibited a decreased fusion/fission ratio compared to control cells, whereas the ratio of fusion/fission was unchanged in cells expressing *Slmo2^RNAi1^* (S8F Fig).

## Discussion

Organelle membranes in eukaryotic cells are composed primarily of phospholipids of different varieties. Even minor changes in lipid composition can dramatically affect cellular metabolism

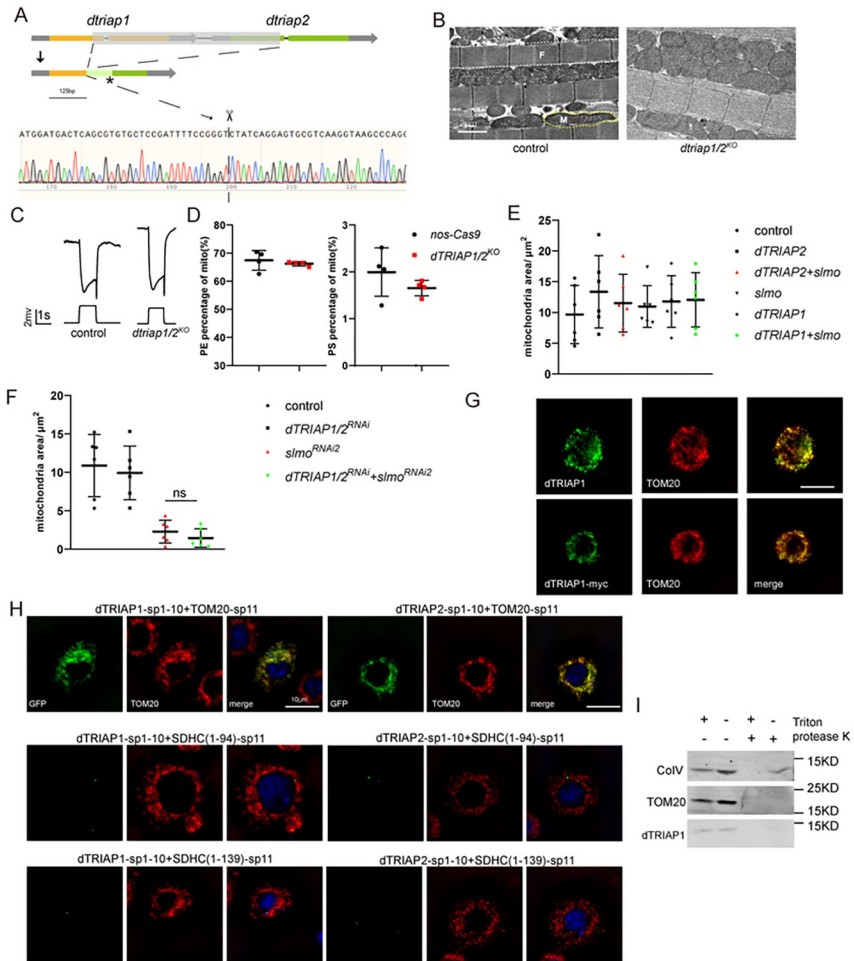

**Fig 6. SLMO functions independently of dTRIAP1 and dTRIAP2.** The *dtriap1 and dtriap2* loci and mutation sites associated with *dtriap1/2^KO^*. The asterisk indicates a stop codon. (B–D) Knocking out *dtriap1* and *dtriap2* does not affect (B) mitochondrial morphology, (C) neuronal function, or (D) PE/PS level of mitochondria. (B) TEM sections of muscles from wild-type (*nos-Cas9*) and *dtriap1/2^KO^* flies show similar mitochondrial size. M, mitochondria; F, myofibrils. Scale bar, 2 μm. (C) ERG responses of wild-type and *dtriap1/2^KO^* flies are shown. (D) Lipidomic analysis of mitochondrial phospholipid levels of control (*nos-Cas9*), *dtriap1/2^ko^* muscles. Individual phospholipid levels were calculated in molar fractions of total phospholipids. Mitochondria were isolated from 10 dissected thoraxes per genotype, and 4 replicates were quantified. (E) co-expression of dTRIAP1 and SLMO did not affect mitochondria size. Quantification of mitochondria size in indirect flight muscles from control (*MHC-gal4/UAS-RFP*), *dtriap2* (*MHC-gal4/UAS-dtriap2*), *dtriap2+slmo* (*MHC-gal4/+;UAS-slmo/UAS-dtriap2*), *slmo* (*MHC-gal4/UAS-slmo*), *dtriap1* (*MHC-gal4/UAS-dtriap1*), and *dtriap1+slmo* (*MHC-gal4/+;UAS-slmo/UAS-dtriap1*) flies. All flies were raised for 5 days under 12 h-light/12 h-dark cycles. (F) Knockout of *dtriap1/2* failed to affect mitochondrial morphology in *slmo^RNAi^* flies. Quantification of mitochondria size in flight muscles from control (*MHC-gal4/UAS-GFP*), *dtriap1/2^RNAi^* (*MHC-gal4/UAS-dtriap1/2^RNAi^*), *slmo^RNAi2^* (*MHC-gal4/+;UAS-slmo^RNAi2^/+*), and *dtriap1/2^RNAi^+slmo^RNAi2^* (*MHC-gal4/UAS-dtriap1/2^RNAi^;UAS-slmo^RNAi2^*) flies. Six samples were analyzed for each phenotype. (G–I) dTRIAP1 and dTRIAP2 localize to the OMM. (G) dTRIAP1 is located to mitochondria. S2 cells were transfected with untagged and MYC-tagged dTRIAP1 (dTRIAP1-myc, bottom) and labeled with antibodies against dTRIAP1 (top, green), MYC (bottom, green), and TOM20 (red). Scale bar, 10 μm. (H) S2 cells co-expressing dTRIAP1-sp1-10 or dTRIAP2-sp1-10 with TOM20-sp11, SDHC(1–94)-sp11, or SDHC(1–139)-sp11 were imaged for GFP fluorescence (green); TOM20 (red) was used to visualize mitochondria. Scale bar, 10 μm. (I) Protease K protection assay of dTRIAP1 in S2 cells transfected with dTRIAP1. The data underlying the graphs shown in the figure can be found in S1 Data. ERG, electroretinogram; OMM, outer mitochondrial membrane; PE, phosphatidylethanolamine; PS, phosphatidylserine; TEM, transmission electron microscopy.

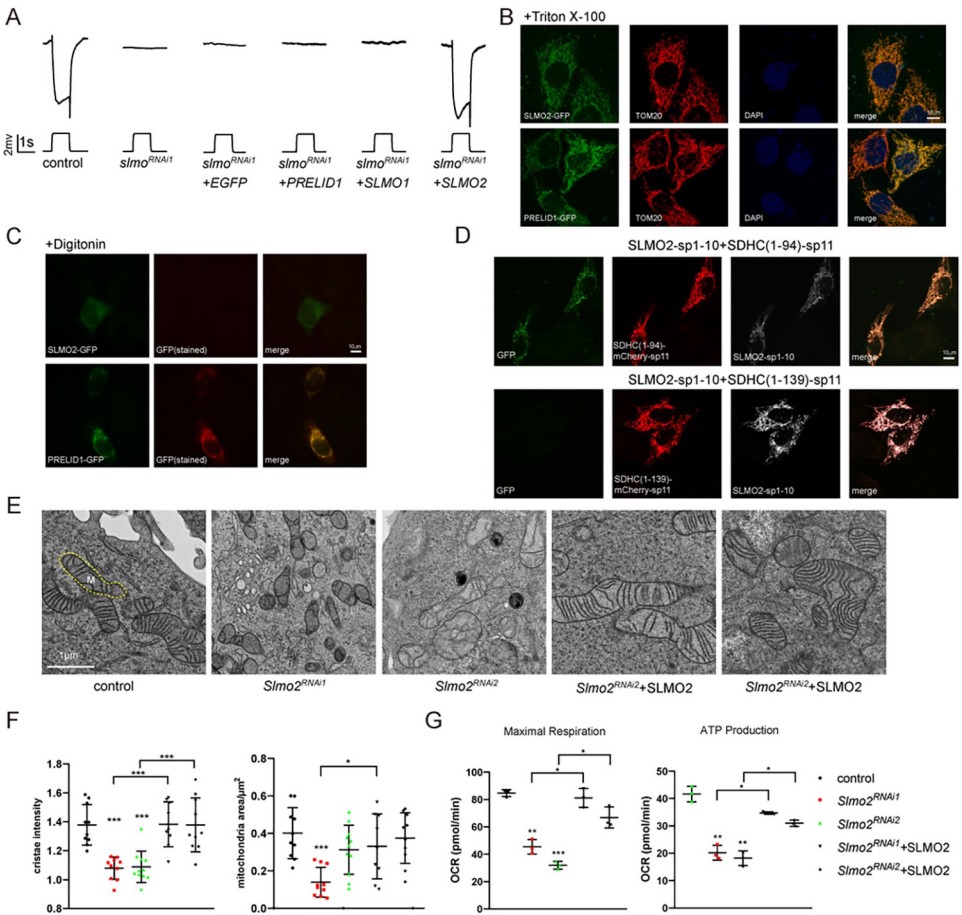

**Fig 7. SLMO function is conserved in mammals.** (A) Expressing SLMO2 restored ERG responses of *slmo* mutant flies. Five-day-old control (*GMR-Gal4/UAS-EGFP^RNAi*), *slmo^RNAi1* (*GMR-Gal4/UAS-slmo^RNAi1*), *slmo^RNAi1+EGFP* (*GMR-Gal4/+;UAS-slmo^RNAi1/UAS-EGFP*), *slmo^RNAi+PRELID1* (*GMR-Gal4/+;UAS-slmo^RNAi1/UAS-PRELID1*), *slmo^RNAi1+SLMO1* (*GMR-Gal4/+;UAS-slmo^RNAi1/UAS-SLMO1*), and *slmo^RNAi1+SLMO2* (*GMR-Gal4/+;UAS-slmo^RNAi1/UAS-SLMO2*) flies were recorded. (B–D) SLMO2 localized to the IMS of mitochondria. Hela cells expressing SLMO2-GFP or PRELID1-GFP were penetrated by (B) triton or (C) digitonin, and GFP was stained (red, in C) or directly observed (green, in C). TOM20 (red, in B) are stained as markers for mitochondria. (D) Hela cells co-expressing SLMO2-sp1-10 (white) with SDHC(1–94)-mCherry-sp11 or SDHC(1–139)-mCherry-sp11 were directly imaged for GFP fluorescence (green) and mCherry fluorescence (red). Scale bar, 10 μm. (E) TEM images show that knocking down *Slmo2* in Hela cells caused mitochondrial fragmentation and cristae loss. Mitochondria were labeled as "m." *Slmo2* was knocked down in Hela cells by transfecting them with pLKO vectors expressing *Slmo2^RNAi1* and *Slmo2^RNAi2* and were rescued by expressing wild-type SLMO2 with corresponding *shRNA*-resistant DNA sequence changes. (F) Mitochondrial size and cristae density were quantified from the mitochondria of more than 20 cells in each group. (G) OCRs in Hela cells transfected with control, *Slmo2^RNAi1*, *Slmo2^RNAi2*, *Slmo2^RNAi1+SLMO2*, and *Slmo2^RNAi2+SLMO2*. ATP production and maximal respiration of the indicated genotype were calculated by normalization of OCR levels to ATP levels. The data underlying the graphs shown in the figure can be found in S1 Data. ATP, adenosine triphosphate; ERG, electroretinogram; IMS, inner membrane space; OCR, oxygen consumption rate; TEM, transmission electron microscopy.

and lead to a variety of diseases including nonalcoholic fatty liver disease (NAFLD) and neurodegenerative diseases [45–47]. Mitochondria are key metabolic organelles and must maintain a specific composition of phospholipids to function properly. Mitochondrial phospholipids are synthesized in the ER and then transported to mitochondria; however, it remains unclear whether phospholipid signals and/or individual phospholipid species regulate mitochondrial morphology and structural organization. Here, we found that knocking down PSS resulted in small mitochondria with reduced function, suggesting that PS trafficking from the ER to

mitochondria may be important for driving mitochondrial biogenesis. In mammalian cells, phosphatidylserine synthase activity is highly enriched in EMCS, where PS production might directly promote PS transport from the ER to mitochondria [48]. Two PSS genes might function redundantly in mammals, since mice lacking either PSS1 or PSS2 are viable, but the double knockout of these 2 genes is lethal [49]. It is reasonable that disruption of both PSS1 and PSS2 may disrupt mitochondrial function and morphology, leading to lethality as in *Drosophila*. Moreover, the accumulation of mitochondrial PS (by knocking down PISD) inhibited the transport of PS from the ER to mitochondria, leading to fragmented mitochondria with reduced function. This is consistent with previous reports indicating that PISD knockdown in the skeletal muscle of adult mice leads to the accumulation of mitochondria with aberrant morphology [19,21]. Additionally, the supplementation of PE has been shown to enhance mitochondrial respiratory capacity in skeletal muscle following exercise training, underscoring the importance of PE in mitochondrial metabolism [25]. These evidence highlights the critical roles of the generation of PE in mitochondria from PS derived from the ER in mitochondrial biogenesis. We show here that knocking down the newly identified *slmo* caused a dramatic accumulation of PS on the OMM by preventing the transporting of PS from the OMM to the IMM. This led to fragmented mitochondria with reduced cristae and disrupted function. Therefore, PS trafficking from the ER to mitochondria and from the OMM to the IMM is essential for maintaining mitochondrial size and cristae structure.

Phospholipid transport between 2 lipid biolayers is energetically unfavorable, and therefore rarely occurs spontaneously [50]. Lipids are generally transferred via vesicle trafficking or non-vesicle transfer by transporting proteins. Utilizing a forward genetic screen, we found that SLMO is involved in PISD-activated mitochondrial PE synthesis and can help compensate for reduced levels of cellular PE in *pect* mutants. In *slmo* mutant cells, mitochondria are fragmented and have fewer cristae, leading to loss of mitochondrial function and eventually cell death. Epistatic analysis revealed that SLMO overexpression suppressed the mitochondrial phenotypes of *pss* mutant muscles and that overexpression of PISD could prevent the degeneration of photoreceptor cells caused by a slight reduction in SLMO to about 55% (S2 Table). Severe SLMO deficiency is truly difficult to compensate by PISD overexpression, mainly because too much PISD can also cause mitochondria damage. This indicated that SLMO functions downstream of PSS and upstream of PISD. We did notice the difference of *pisd* between the retina and thoraxes, knocking down *pisd* in muscle caused decreased mitochondrial size but had no obvious effect in the retina, this may differ from the need for phospholipids of mitochondria in different tissues. The same is true for *pss$^{RNAi}$*. As seen with yeast homologs of SLMO (Ups1 and Ups2), SLMO localized exclusively to the IMS [15,16]. In vitro biochemical analysis confirmed that SLMO specifically transfers PS, similar to SLMO2/Ups2 [15,17]. We further identified 2 conserved amino acids that are critical for SLMO to transport PS. Mutation of Thr93 or Asn150 abolished the PS transport activity of SLMO, and flies with *slmo$^{T93A}$* or *slmo$^{N150A}$* mutations had mitochondrial defects and resulting physiological phenotypes. Although we were unable to determine whether these mutations affected protein folding, levels of expression and localization were unaffected. In summary, these studies establish that SLMO-mediated PS transport is critical to maintaining mitochondrial morphology and function.

Ups1 and Ups2 are both chaperoned by the TRIAP1/Mdm35 complex, which imports and stabilizes both proteins. This is critical for the transport of phospholipids from the OMM to the IMM (Aaltonen and colleagues [15]; Connerth and colleagues [16]; Kay and colleagues [41]; Potting and colleagues [13,18]). However, unlike yeast Mdm35, dTRIAP1 and dTRIAP2 both localize to the OMM, whereas SLMO is found in the IMS. The *dtriap1/2* double mutants had normal mitochondrial morphology, lipid content, and cellular function, confirming that

SLMO functions independently of dTRIAP1/2. Moreover, the complete loss of dTRIAP1/2 did not affect the function or stability of SLMO, and knocking down *dtriap1*/2 did not aggravate the mitochondrial defects of *slmo^RNAi^*. These results suggest that the findings may not align with those observed in yeast studies. In yeast, Mdm35 protects Ups2 against proteolysis, and cells lacking Mdm35 lead to the degradation of Ups2 by Yme1, a subunit of the ATP-dependent i-AAA protease active in the intermembrane space [13,51]. However, in *Drosophila*, loss of dTRIAP1/2 does not reduce SLMO levels, consistent with different localization of dTRIAP1/2 and SLMO, supporting functions of SLMO independent of dTRIAP1/2. Although SLMO in flies might not rely on dTRIAP for its stability, it is interesting to investigate whether Yme1 plays a similar role in controlling SLMO levels to maintain phospholipid homeostasis. The IMM is subdivided by cristae junctions (CJs) into 2 structurally and functionally distinct domains, the CM and the inner boundary membrane (IBM) [52]. IMS and IMM proteins localize to the CM and IBM regions to function properly. For example, the ATPase and electron transport chain subunits are enriched in the CM region to ensure oxidative phosphorylation and ATP production. In contrast, MICU1 localizes to the IBM where it facilitates rapid uptake of $Ca^{2+}$ by mitochondria when cytosolic levels of $Ca^{2+}$ are elevated [39]. SLMO localizes exclusively to the IMS regions, which is where phospholipids are transferred between the OMM and IMM.

Mitochondria are highly dynamic organelles and maintain their shape, organization, and size through mitochondrial dynamics that involve organelle fusion and fission [53]. Ups1 regulates mitochondrial morphology by processing the dynamin-related GTPase Mgm1p, which is the yeast homology of Opa1 required for mitochondrial fusion [14]. In *Slmo2* knocked-down cells, mitochondria were fragmented, distorted, and empty, but levels of mitochondrial fusion and fission were not affected. Moreover, we present genetic evidence that induction of mitofusion or mitofission had little effect on the mitochondrial structural organization in *slmo* mutant muscle cells. Therefore, SLMO is involved in shaping mitochondrial morphology independent of fission/fusion events.

The topological structure of transmembrane lipid transfer proteins determines the direction of phospholipid transport. However, SLMO lacks a transmembrane helix and therefore may freely shuttle lipids between the inner and outer membranes, mediating non-directional transport. Unidirectional PS trafficking between the OMM and IMM may be maintained by the PS concentration gradient. Supporting this, an in vitro assay was used to show that PS transfer is accelerated by elevated levels of PS in the donor membrane [54,55]. Although PS accounts for only 5% of total phospholipids in the ER [56], in situ synthesis of phospholipids in the ER makes the local PS concentration in the ER far more than seen in the OMM at ERMCS. This gradient promotes the translocation of PS to mitochondria and disruption of this gradient (by inhibiting PSS) reduces mitochondrial levels of PS and PE, resulting in smaller mitochondria. PS that reaches the OMM is captured by SLMO and delivered to the IMM where PS is rapidly converted to PE by PISD. This helps establish the PS gradient between the OMM and IMM [55,57]. Remarkably, in both yeast and mammalian cells deletion of mitochondrial PISD alters mitochondrial morphology and impairs oxidative phosphorylation, suggesting that mitochondrial PE promotes the formation of cristae and mitochondrial fusion [21,22,24,58]. However, in yeast, targeting PISD to the OMM or ER prevents loss of mitochondrial PE, but fails to rescue the respiratory defects of complex III, challenging the notion that PE plays a role in maintaining mitochondrial organization [24]. Alternatively, knocking down PISD results in the accumulation of mitochondrial PS and reduces PS influx, thus reducing mitochondrial size. Similarly, disrupting the SLMO-mediated transfer of PS from the OMM to the IMM resulted in the accumulation of PS in the OMM. This blocked the

transport of PS from the ER to mitochondria, leading to smaller mitochondria, less oxidative phosphorylation, and ultimately cell dysfunction and death.

In summary, our findings demonstrate that PS trafficking from the ER to mitochondria and between mitochondrial membranes is required for maintaining mitochondrial morphology. Disrupting PS trafficking results in fragmented mitochondria, less mitochondrial oxidative phosphorylation, cellular dysfunction, and ultimately cell death.

## Methods

### Fly stocks

The following stocks were obtained from the Bloomington Stock Center: *w\*;P[long-GMR-GAL4]2, y1 sc\* v1 sev21; P[VALIUM20-EGFP.shRNA.4]attP2, w^1118, y1 sc\* v1 sev21; P {TRiP.GL01830}attP40* (*pis^RNAi*), *y1 sc\* v1 sev21; P{TRiP.HMC04893}attP40* (*bbc^RNAi*), *M(vas-int.Dm)ZH-2A;M(3xP3-RFP.attP)ZH-86Fb, M(vas-int.Dm)ZH-2A;M{3xP3-RFP.attP}ZH-51C, nos-Cas9.* The *pcyt1^RNAi*, *serca^RNAi*, *pss^RNAi*, and *pisd^RNAi* lines were obtained from the Tsinghua stock center. The *ey-flp Rh1::GFP;GMR-hid CL FRT40A/Cyo hs-hid, MHC-GAL4 UAS-mitoGFP, UAS-EGFP, trp-pisd, UAS-pss* were maintained in the laboratory of T. Wang [29]. Flies were maintained in 12 h light/12 h dark cycles with 2,000 lux illumination at 25°C. For animal studies, no randomization and no blinding were used.

### Generation of plasmid constructs and transgenic flies

*slmo^RNAi* lines were generated using the V20 system as described [59,60], with the following short hairpin RNA sequences: *slmo^RNAi*, 5′- ACGATGGACATGGACAGCAACATGT-3′; *slmo^RNAi5*, 5′-GGGCTGTTCACTAGTGAATTA-3′. Annealed oligo pairs were cloned into the *VALIUM20* vector containing *w* instead of *v*. The final constructs were injected into *M(vas-int.Dm)ZH-2A;M(3xP3-RFP.attP)ZH-86Fb* or *M(vas-int.Dm)ZH-2A;M{3xP3-RFP.attP}ZH-51C* embryos, and transformants were identified based on eye color. Multiple *pect^RNAi* lines were generated using the same method with the following primers: *pect^RNAi1*: 5′-GCAATGGG-TATTCGGACAAGC-3′; *pect^RNAi2*: 5′- CGGCCAACCGAACCCTACTATTTCA-3′; *pect^R-NAi2w*: 5′-TCGTCCGGCTGAAATGTCTACATAA-3′; *pect^RNAi3*: 5′-GCTCTCACGGTACTCG AAATT-3′; *pect^RNAi5*: 5′-CATTTCCTGCTGTCGAACTTGTTTA-3′; *pect^RNAi6*: 5′- CCCAGT TTGTCAATGAGGTAGTAAT-3′.

*slmo^RNAi1*, *slmo^RNAi2*, and *pss^RNAi1-2* lines were generated using the *pNP* system as described, replacing the marker *v* with *w* [32]. The short hairpin RNA sequences of *slmo^RNAi1* and *slmo^R-NAi2* are 5′-GGACATCGGAGCACATATTCA-3′ and 5′-GCACATATTCAACCACCCGTG-3′. The *pss^RNAi1-2* combined 2 short hairpin sequences (5′-GGAGCATATTCTACTGGATTG-3′ and 5′-GGGATCATCCCGCCTACAAAT-3′). The constructs were injected into *M(vas-int. Dm)ZH-2A;M(3xP3-RFP.attP)ZH-86Fb* or *M(vas-int.Dm)ZH-2A;M{3xP3-RFP.attP}ZH-51C* embryos and transformants were identified based on eye pigmentation.

The *slmo* cDNA was amplified from the cDNA clones GH14384, (*Drosophila* Genomic Resource Center). The SLMO1, SLMO2, and PRELID1 cDNAs were amplified from the human cDNA clone IOH10766, IOH9792, and IOH4636, respectively, which were obtained from the Ultimate ORF Clones (Invitrogen). To express GFP-tagged SLMO under the control of the *ubi* or *DA* promoter, *slmo* cDNAs were first ligated with an *EGFP* sequence and subcloned into the *ubi-attB* or *DA-attB* vectors between the Acc65I and XbaI sites. The *slmo^RNAi1*-resistant forms of SLMO were generated via site-directed mutagenesis from *slmo* cDNA using 5′-ATGAAAATCTGGACCTCAGAACATATCTTTAACCACCCGTGGGA-3′ primer. The *slmo^T93A* and *slmo^N150A* were generated via site-directed mutagenesis from the *slmo^RNAi1*-resistant form of *slmo* using 5′-ATGAAAATCTGGACCTCAGAACA-3′ and 5′-

CAGATGGTGCTGAAGGCTAACAA-3′, respectively. To express SLMO1, SLMO2, and PRE-LID1 in flies, cDNAs were synthesized with an optimized codon for protein expression and were subcloned into the *UAST-attB* vector between the Acc65I and XbaI sites. These constructs were injected into *M(vas-int.Dm)ZH-2A;M(3xP3-RFP.attP)ZH-86Fb* embryos, and transformants were identified based on eye color.

## Generation of *slmo¹*, *dtriap1/2^KO*, and *slmo-HA* knock-in flies

The *slmo¹* mutant was generated using the CRISPR/Cas9 system [61,62]. Briefly, a guide RNA targeting the *slmo* locus was designed (sgRNA: CGCCAGCGAACGCTCCACGG), and cloned into the *U6b-sgRNA-short* vector. The plasmid was injected into *nos-Cas9* embryos, and F1 progeny were screened by PCR and DNA sequencing to identify the *slmo¹* mutation using the following primers: forward primer 5′-CTCGGCTGCCAATTAAGGAT-3′ and reverse primer 5′-GAGCGATGCCCTTGACCT-3′ as described in S2B Fig. The *slmo-HA* knock-in line was generated as shown in S4G Fig. Briefly, a single sgDNA sequence (5′- GACAGTG-CAAGGTCGCGCGG-3′) was designed and cloned into the *U6b-sgRNA-short* vector. To generate a donor construct, the 3XHA DNA sequence replaced the stop code of the *slmo* coding region (870 bp upstream and 708 bp downstream of the stop code) and was subcloned into a donor vector. The 2 plasmids were co-injected into the embryos of *nos-Cas9* flies. The *slmo-HA* knock-in flies were identified by PCR of genomic DNA using the following primers: 5′-TGCTCTACTATGAGCCGCAT-3′ and 5′-AGGATGCTGGAGCTGCGTAA-3′ and verified by genomic DNA sequencing. To generate *dtriap1/2^KO* mutation, 2 pairs of guide DNAs targeting the *CG30108* (*dtriap1*) and *CG30109* (*dtriap2*) loci were designed (sgRNA1: 5′-TGAC GCATTCCTGATAAACC-3′, sgRNA2: 5′-TAAGGGCCAGACGGATGACT-3′) and cloned into the *U6b-sgRNA-short* vector. The plasmids were injected into the embryos of nos-Cas9 flies, and deletions were identified by PCR and DNA sequencing using the following primers: forward primer1 5′-AAACAAATGAGCAGCGTTGG-3′ and reverse primer1 5′-AGCTGCT TCTGGTACTTGGT-3′. The *slmo¹*, *dtriap1/2^KO*, and *slmo-HA* flies were backcrossed to wild-type flies (*w^1118*) for 2 generations before preforming experiments [63].

## Electroretinogram (ERG) recordings

ERG recordings were performed as described [64]. Two glass microelectrodes filled with Ringer's solution were inserted into small drops of electrode cream (PARKER LABORATORIES) that were placed onto the surface of the compound eye and the thorax. A Newport light projector (model 765) was used for stimulation. ERG signals were amplified with a Warner electrometer IE-210 and recorded with a MacLab/4s analog-to-digital converter and the Clampex 10.2 program (Molecular Devices, San Jose, California, United States of America). All recordings were carried out at room temperature.

## Transmission electron microscopy (TEM)

TEM experiments for fly tissues were performed as described [64]. Briefly, adult heads or thoraxes were dissected and fixed in a solution with 4% paraformaldehyde and 2.5% glutaraldehyde, followed by a series of dehydration in ethanol. Samples were infiltrated and embedded in Eponate 12 (Ted Pella), and approximately 80 nm thin sections were prepared at a depth of 30 to 40 μm. The sections were stained with uranyl acetate and lead-citrate (Ted Pella) and examined using a JEM-1400 transmission electron microscope (JEOL, Tokyo, Japan) equipped with a Gatan CCD (4k × 3.7k pixels, USA).

For TEM of Hela cells, cells were seeded onto a 12-well plate and then frozen immediately in a high-pressure freezer. The samples were transferred into 2 ml polypropylene screw cap

microtubes (Cat. 81–0204, Biologix) with 1 ml of frozen acetone containing 5% ddH$_2$O +1% OsO$_4$+0.1% UA. The tubes were transferred into the specimen chamber of a Leica AFS2 freeze-substitution machine for freeze-substitution processing. The samples were processed by the following FSF-based oxidization, fixation, and rehydration. Acetone was changed 3 times (15 min intervals) at room temperature. The freeze-substituted fixed specimens were infiltrated with SPI-Pon 812 resin (SPI Inc.). The specimens were embedded in a flat-tip BEEM embedding capsule (Tedpella Inc.) and polymerized at 60˚C for 24 h before 90 nm sections were prepared with a Leica UC7 ultramicrotome.

## Cell culture, transfection, and immunofluorescence

The *slmo* cDNA was cloned into the *pIB-c-GFP* vector, and *Drosophila melanogaster* Schneider 2 (S2) cells cultured in 6-wells plate in Schneider's medium with 10% fetal bovine serum (Gibco, Carlsbad, California, USA) were transfected by adding 5 μg plasmid in 100 μl NaCl (150 mM) mixed with 2 μl VigoFect (Vigorous Biotechnology, Beijing, China) into the culture media. Twenty-four hours after transfection, cells were suspended and adhered to a ConA coverslip for 1 h, followed by fixing with 4% PFA in PBS for 30 min at room temperature. Cells on a coverslip were incubated with primary antibodies against Tom20 (rat, 1:100) [65], GFP (rabbit, 1:200, Invitrogen), Cnx99 (mouse, 1:100, Developmental Studies Hybridoma Bank), or HA (mouse, 1:200, Santa Cruz). Secondary antibodies against rat Alexa Fluor 568, rabbit Alexa Fluor 488, and mouse Alexa Fluor 647 were used (1:400, Invitrogen). Images were captured with a Nikon A1-R confocal microscope (Nikon, Tokyo, Japan). Acquired images were processed using Photoshop CC 2017 and NIS-Elements AR Analysis 5.20.00.

The split-GFP experiments were performed as described [37]. Referring to their strategy, to construct *slmo-spGFP1-10*, *dTriap1-spGFP1-10*, *dTriap2-spGFP1-10*, and *micu1-spGFP1-10*, the cDNAs were amplified and linked to a *spGFP1-10* DNA into *pIB* vectors. To construct *sdhc (1–94)-spGFP11*, and *sdhc(1–139)-spGFP11*, the cDNA fragment of *sdhc* were amplified and linked to 2×*spGFP11* into *pIB*. S2 cells cultured in a 6-well plate were transfected with a combination of *spGFP1-10* and *spGFP11* constructs, and fluorescence was imaged with a Nikon A1-R confocal microscope.

To express SLMO2, PRELID1, and SLMO1 in Hela cells, *SLMO2*, *PRELID1*, and *SLMO1* cDNAs were cloned in the *pCDNA3-c-GFP* vector, and Hela cells were transfected with VigoFect (Vigorous Biotechnology, Beijing, China) by adding 1 μg plasmid per well. Cells on a coverslip were incubated with primary antibodies against Tom20 (Rb, 1:100, Abclone) and ATP5a (mouse, 1:200, Abcam). The split-GFP experiments in Hela cells were performed as in S2 cells, except *micu1-spGFP1-10* was cloned using human *Micu1* cDNA.

Hela cells with stable expression of *shRNA* to *SLMO2* were generated by lentiviral *shRNAs* from Sigma-Aldrich Advanced Genomics using pLKO.1 vector SHC001, TRCN0000121980 (*SLMO2$^{RNAi1}$*) and TRCN0000122136 (*SLMO2$^{RNAi2}$*). HEK293T cells were transfected by individually adding these 3 vectors with virus assemble construct *pMD2.G* (Addgene#12260) and *psPAX2* (Addgene#12259) in a ratio below 1 μg *pLKO.1 shRNA* plasmid; 750 ng *psPAX2* packaging plasmid; 250 ng *pMD2.G* envelope plasmid. After 48 h post-transfection, the medium containing lentiviral particle solution was filtered through a 0.45 μm filter to remove the cells and transferred to Hela cells. After incubation for 2 days, puromycin was added to the medium to the final concentration of 10 μg/ml. The stable cell lines were maintained in 1 μg/ml puromycin. To generate *Slmo$^{RNAi}$* rescue cell lines, the synthesized *SLMO* cDNA, which is resistant to *shRNA*, was subcloned into *pLVX-CMV-IRES-neoR* plasmid and transfected *Slmo2$^{RNAi1}$* and *Slmo$^{RNAi2}$* cells.

The Mfn1 knockout cell line was generated using CRISPR/Cas9 system [66]. Primers against the gRNA targeting site were annealed and ligated into the pX458 vector. The plasmid

was transiently transfected into HeLa using JetPRIME (Polyplus 101000015), and 48 h after transfection, cells were trypsinized and GFP-positive cells were sorted into 96-well plates as single cells using a BD FACSAria III flow cytometer (1 cell per well). Single clones were grown for 2 to 3 weeks and expanded. Knockout clones were examined by immunoblotting against MFN1 (Rabbit D6E2S, Cell Signaling Technology 14739S) and DNA sequencing (MFN1 sgRNA primers: MFN1-e3-55rev-F: caccgTTATATGGCCAATCCCACTA; MFN1-e3-55rev-R: aaacTAGTGGGATTGGCCATATAAc).

### RNA interference and PS reporting system in S2 cells

Double-stain RNAs (dsRNAs) against *slmo*, *pss*, *pisd*, and *BFP* were synthesized in vitro and transfected into S2 cells as described [67]. Briefly, DNA primers containing T7 promoter sequence were designed to amplify approximately 500 bp of cDNA (*slmo*: 5′-TAATACGACT CACTATAGGGAGACCACTCGGAGCACATATTCAACCA-3′/5′-TAATACGACTCACTA TAGGGAGACCACCTACGTAATGTGCATCGCCT-3′; *BFP*: 5′-TAATACGACTCACTAT AGGGAGACCACATGAGCGAGCTGATTAAGGA-3′/5′-TAATACGACTCACTATAGGG AGACCACATTAAGCTTGTGCCCCAG-3′; *pss*: 5′-TAATACGACTCACTATAGGGAGAC CACATGAAGAAGCGCACTAATTCAC-3′/5′-TAATACGACTCACTATAGGGAGACC ACCCCATGTGACGAATGAGGAT-3′; *pisd*: 5′-TAATACGACTCACTATAGGGAGACC ACGAGCGAGGCCATGTATCCG-3′/5′-TAATACGACTCACTATAGGGAGACCACT TGGGGCTCACGGACAGCA-3′). Long dsRNAs were transcribed in vitro using MEGA shortscriptT7 transcription system (Invitrogen, AM1354) as described using the universal T7 primer according to the protocol of MEGA shortscript. The dsRNAs were added to the culture medium at a final concentration of 3 μg/μl 3 days before analysis.

To express the C-terminal GFP-tagged C1C2 domain of mouse lactadherin in S2 cells, the cDNA encoding mouse lactadherin C1C2 domain was synthesized and subcloned to the pIB-cGFP vector. The S2 cells were transfected with *LactC1C2-GFP* using a VigoFect reagent.

### Live imaging of Hela cells

To count the fusion and fission rate of mitochondria, Hela cells were transfected with Cox8-GFP to label mitochondria. Images were acquired using a Nikon A1-R coupled with an Andor Dragonfly 500 confocal spinning disk system using the 60× objective. For live-cell imaging analysis, time-lapse videos were acquired for 5 min with an image captured every 10 s. A region of 625 μm$^2$ per cell was cropped using Fiji software, and the number of fission and fusion events was manually counted.

### Immunohistochemistry

A variety of fly tissues including testis, muscles, and retina were dissected in PBS buffer. The tissues were then fixed with 4% PFA for 30 min at room temperature (for muscle the concentration of PFA should reduce to 2%) and penetrated by 0.1% Triton or 0.5 μg/ml Digitonin (Sigma-Aldrich, D141). Immunolabeling was performed with Tom20 (rat, 1:100) [65], HA (mouse, 1:200, Santa Cruz Biotechnology), and GFP (rabbit, 1:200, Invitrogen) as primary antibodies. Goat anti-rabbit lgG conjugated to Alexa 488 or Alexa 568 (1:500, Thermo Fisher, USA), goat anti-mouse lgG conjugated to Alexa 488 (1:500, Thermo Fisher, USA), and goat anti-rat lgG conjugated to Alexa 647 (1:500, Thermo Fisher, USA) were used as secondary antibodies. The images were recorded with a Nikon A1-R or Nikon A1-SIM confocal microscope.

## Lipid extraction and mass spectrometric analysis

Mitochondria were extracted from dissected *Drosophila* thorax using a Cell Mitochondria Isolation Kit (Beyotime Biotechnology, C3601) or through fluorescence-activated cell sorting (FACS) [68], following tissue homogenization using a Dounce homogenizer (KONTES). Lipids were extracted from muscle mitochondria as described [69]. Briefly, 225 μl MeOH and 750 μl methyl tert-butyl ether (MTBE) were used to extract phospholipid from ground tissue or mitochondria. Ten microliters of lipidomic analytical standard (Avanti, 330707) was added to each sample before extraction.

Polar lipids were analyzed on an Exion UPLC system coupled with a triple quadrupole/ion trap mass spectrometer (QTRAP 6500 PLUS, Sciex). PC-d31 (16:0/18:1), PE-d31 (16:0/18:1), PS-d31 (16:0/18:1), PI-d31 (16:0/18:1), PA-d31 (16:0/18:1), PA (17:0/17:0), PG-d31 (16:0/18:1), LPC-17:0, LPE-17:1, LPS-17:1, LPI-17:1 were obtained from Avanti Polar Lipids (Alabaster, Alabama, USA) as standard samples. Glycerol lipids triacylglycerides (TAG) and triacylglycerides (DAG) were analyzed using a modified protocol of reverse-phase HPLC/ESI/MS described previously [70]. DAG species were quantified using 4ME 16:0 Diether DG as an internal standard (Avanti Polar Lipids, Alabaster, Alabama, USA). Separation of individual lipid classes of polar lipids by NP-HPLC was carried out using a Phenomenex Luna 3 μ-silica column (i.d. 150 × 2.0 mm) with the following conditions: mobile phase A (chloroform: methanol: ammonium hydroxide, 89.5:10:0.5) and mobile phase B (chloroform: methanol: ammonium hydroxide: water, 55:39:0.5:5.5). The initial mobile phase proportion was 95% (A)/5% (B), which was held for 5 min. Mobile phase A was then linearly decreased to 60% in 7 min, which was held for 4 min, and was further decreased to 30%, which was held for 15 min. Finally, the initial mobile phase conditions were restored and held for 5 min. Data were acquired in multiple reaction monitoring (MRM) in a combined workflow for polar lipids analysis. The source parameters are as follows: curtain gas, 20; ion spray voltage, 5500 V; temperature, 400˚C; ion source gas 1, 35; ion source gas 2, 35. MS profiles were recorded under both positive and negative modes in separate runs (resolution 60,000), and mass accuracy of less than 2 ppm was obtained throughout the analytical runs. The mole fraction of each lipid was normalized to the mole fraction of total polar lipids.

## Quantitative real-time PCR

RNAi efficiencies of RNAi flies driven by *GMR-Gal4* were analyzed by real-time polymerase chain reaction (qPCR). RNA was isolated from dissected adult fly retinas expressing the individual *shRNAs* (*GMR-Gal4/UAS-RNAi*) using Trizol (Invitrogen). Total RNA was treated with TURBO DNase (Thermo Fisher), and 500 ng RNA was subjected to reverse transcription using the PrimeScript RT Master Mix (Takara). Quantitative PCR was performed using TB Green *Premix Ex Taq* II (Takara), and results were analyzed with a CFX96 Real-Time PCR Detection System (Bio-Rad). Primers used for qPCR are as follows: *slmo* forward primer: 5′-TCGGAGCACATATTCAACCACC-3′; *slmo* reverse primer: 5′-GATGCGGCTCATAGTAGAGCA-3′; *pss* forward primer: 5′-GTCCTGTGGCTGACTGTCCA-3′; *pss* reverse primer: 5′-ATCCACCATAGACATGAACAGACG-3′; *pisd* forward primer: 5′-ATCTCAGACAGGTGGTGCAG-3′; *pisd* reverse primer: 5′-TGGAGCTCGGACGCTGTC-3′; *SLMO2* forward primer: 5′-GCAAGTACGATATCCTCAAATGCTAG-3′; *SLMO2* reverse primer: 5′-TCACTTCTCTGCAAACGCTG-3′; *RPL32* forward primer: 5′-ATCGGTTACGGATCGAACAA-3′; *RPL32* reverse primer: 5′-GACAATCTCCTTGCGCTTCT-3′; *GAPDH* forward primer: 5′-CAATGCCTCCTGCACCAC-3′; *GAPDH* reverse primer: 5′-GTTGGCAGTGGGGACAC-3′. The knock-down efficiencies of RNAi lines are described in S2 Table.

## Protein purification

The *slmo*, *slmo^{T93A}*, and *slmo^{N150A}* cDNA were subcloned into the *pMAL-c6T* Vector (New England Biolabs, E8201S) between the BamHI and EcoRI sites, and the plasmid was transformed into BL21 (DE3) competent cells (TransGen Biotech, CD601-02). Protein expression was induced with 0.1 mM IPTG, and cells were harvested and resuspended in isolation/wash buffer (20 mM HEPES-KOH, 100 mM NaCl, 0.5 M EDTA, 25 Mm imidazole) supplemented with complete EDTA-free protease inhibitor cocktail (Sigma). Cells were lysed by freeze-thaw followed by sonication. After loading the diluted crude extract into the HisPur Ni-NTA Resin (Thermo Scientific, 88221), the resin was washed with 12 column volumes of wash Buffer. Elution of protein was generated by adding elution buffer (20 mM HEPES-KOH, 100 mM NaCl, 0.5 M EDTA, 250 mM imidazole) to the resin.

## Lipid strip

Membrane Lipid Strips (Echelon Biosciences Incorporated, P-6002) were used to assess the specificity of SLMO for binding with various phospholipids. The membrane was blocked with 5 ml of TBS (20 mM Tris-HCl (pH 7.5), 150 mM NaCl) with 0.1% (w/v) ovalbumin (Sigma, A5253), and gently shaken for 1 h at room temperature. MBP-SLMO, MBP, or PI (4,5) P2 GripTM protein was added to a final concentration of 5 μg/ml. After incubated at room temperature for 1 h, the membrane was washed 3 times with 5 ml of TBST (with 0.05% Tween20), followed by incubation with mouse-anti-MBP (New England Biolabs, E8032S, 1:10,000) or anti-GST antibodies (Santa Cruz, D2808, 1:2,000) in TBST for 1 h at room temperature with gentle shaking. The HRP-labeled Goat Anti-Mouse IgG (Beyotime Biotechnology, A0216, 1:5,000) was used as secondary, and signals were detected by Electrochemiluminescence (ECL) reaction.

## Lipid flotation assay

MBP-SLMO, MBP, MBP-SLMO(T93A), MBP-SLMO(N150A) (5 μm) were incubated with liposomes (PC: PS = 96:4, PC: PE = 56:44) in 100 μl buffer A (HEPES/KOH, pH 7.5, 150 mM NaCl, and 1 mM EDTA) at 25°C for 10 min. After incubation, the sample was mixed with 100 μl of 60% sucrose in buffer A, put into an ultracentrifuge tube, and overlaid with 3.35 ml 30% sucrose in A, 1 ml buffer F10 (10% sucrose in F), and 500 μl buffer A. Tubes were centrifuged at 200,000 g for 1.5 h. Two fractions of 500 μl were collected from the top and bottom respectively, and proteins in the fractions were precipitated by TCA.

## Lipid transfer assay

Lipid transfer assay was performed as described [15–17] with some modifications. Phospholipids including 18:1 (Δ9-Cis) PC, 18:1 PS (DOPS), 18:1 (Δ9-Cis) PE (DOPE), 18:1–12:0 NBD PS, 18:1 Liss Rhod PE, and 18:1 Cardiolipin were obtained from Avanti Polar Lipids. Lipid stock in chloroform solutions was mixed at the desired molar ratio (Donor liposome: PC: PE: Rhod-PE: NBD-PS = 56: 40: 3: 1, 10 mM; acceptor liposome: PC: PE: CL = 55: 40: 5, 10 mM), and the solvent was evaporated under a flow of argon. The lipid film was hydrated in 300 μl wash buffer. Ten freeze–thaw cycles in liquid nitrogen were carried out, and the thawed solution was extruded 20 times through a 0.1-μm filter using a mini-extruder (Avanti Polar Lipids) to form liposomes, and 125 μm donor (final) and 50 μm acceptor (final) were added into the wash buffer for 200 μl final volume. After protein in elution buffer (diluted into 0.5 μm) was added, dequenching of NBD fluorescence upon addition of accepter liposomes was monitored in SpectraMax Paradigm microplate reader.

## Docking, MD simulation, and binding free energy calculation

The docking of small molecules and proteins was scored by the Molecular Operating Environment (MOE ver. 2019.0102). *Drosophila* SLMO was prepared using the QuickPrep panel with default parameters to delete all water molecules and complete the missing atom. After preparation, a set finder was used to find the binding sites. The placement method used Triangle Matcher [71], and the protein was set to rigid for refinement. London dG for placement and GBVI/WSA dG [72] for refinement were used to estimate the free energy of binding of the ligand from a given pose and output the final score. Each system exported the top 10 highest-scored poses from 100 poses.

The MD simulations of PS and SLMO were conducted using the Amber 19 package. The initial conformation was the lowest score pose in former docking. The force field of small molecules was calculated by an antechamber with the general AMBER force field (GAFF) [73]. The protein was processed by the ff14SB force field [74]. The systems were solvated in a TIP3P water box and performed for the energy minimization under harmonic restraints with a 200 kcal mol$^{-1}$Å$^{-2}$ for 2,500 steps and another 5,000 steps without any restraints. The whole system was then gradually heated from 0 to 297 K and equilibrated at 297 K for 0.5 ns. Finally, MD simulations used an NTP ensemble at 297 K for 20 ns, long enough for the system to reach and simulate at the steady state. The cutoff distance for van der Waals (vdW) interactions was set to 12 Å. The particle mesh Ewald method was employed to calculate the electrostatic interaction with a real space cutoff which equals 10 Å. The integration time step was 2.0 fs and the peptide coordinates were saved every 2 ps after the system reached the steady state.

The last 10 ns trajectories were used to perform MM/GBSA analysis to calculate free energy by amber. MM/GBSA method calculates the free energy difference between 2 states which most often represent the bound and unbound state of 2 solvated molecules. The GB model we used is the GB$^{OBC1}$ model provided by Amber20. It should be noted that in AMBER, the GB$^{OBC1}$ model was parameterized for Bondi radii [75]. The non-polar contribution was determined based on solvent-accessible surface area (SASA) with the LCPO method [76].

## Western blotting

Western blotting was performed by homogenizing dissected fly tissues or mitochondria from S2 cells in an SDS sample buffer with a Pellet Pestle (Kimble/Kontes). For protease K protection assay, S2 cells transfected with *pIB-slmo-GFP*, *pIB-sdhc(1–94)- mCherry* and *pIB-mito-GFP* (*Cox8-Sod2-GFP*) were suspended in homogenization buffer (20 mM HEPES/KOH (pH = 7.4), 1 mM EDTA, 210 Mm Mannitol, 70 mM Sucrose, 0.5% BSA) and incubated on ice for 15 min. Then, cells were homogenized by Pellet Pestle, followed by centrifugation at 1,000g for 5 min. The supernatant was divided into 4 parts and centrifuged at 3,500g for 10 min to collect mitochondria. Pellets with mitochondria were re-suspended in 40 μl homogenization buffer, homogenization buffer+0.1% TritonX100, homogenization buffer+50μg protease K, and homogenization buffer+0.1% TritonX100+50μg protease K in 37°C for 30 min. Then, the digestion was stopped by PMSF for 15 min on ice. The proteins were fractionated by SDS-PAGE and transferred to Immobilon-P transfer membranes (Millipore) in Tris-glycine buffer. The blots were probed with Tom20 (rat, 1:1,000) [65] and CoIV (Rb,1:2,000) [65], GFP (Rb,1:1,000, Torrey Pines Biolabs), mCherry (Rb,1:1,000, biovision), β-Actin (mouse, 1:1,000, Santa Cruz Biotechnology) followed by IRDye 680 goat anti-Rabbit IgG (LI-COR) and IRDye 800 goat anti-Mouse IgG (LI-COR) as the secondary antibodies. The signals were detected with the Odyssey infrared imaging system (LI-COR).

## Climbing assay in adult *Drosophila*

The climbing assay was performed according to previous studies [77] with some modifications. Flies of either sex were collected in vials with standard corn meal media in batches of more than 10 and aged for 3 days. Flies were transferred into the cylinder without anaesthetization by tapping the vials. Flies were incubated for 3 to 4 min at room temperature to allow for environmental adaptation, followed by tapping flies to the bottom of the vial. The percentage of flies that crossed the 13 cm mark within 30 s was recorded. This was repeated 2 more times with the same set of flies with an interval of 3 to 4 min in between trials to calculate the average climbing ability for each batch of flies. A minimum of 3 batches of 10 flies were tested per genotype.

## Measurement of oxygen consumption rate (OCR)

The OCR was measured using an Agilent Seahorse XFe96 extracellular flux analyzer (Seahorse Bioscience, North Billerica, Massachusetts, USA). Cells were seeded into 96-well XF-PS plates at a density of 8,000 cells per well in 80 μl growth medium (DMEM medium supplemented with 10% fetal bovine serum, 1% penicillin, and streptomycin). Cells were plated with at least 3 replicate wells for each treatment group, followed by overnight culture at 37˚C in a CO2 incubator. The XF extracellular flux sensory cartridge was hydrated in a non-CO2, 37˚C incubator for 12 to 18 h. On the day of measurement, cells were washed twice with assay medium composed of Seahorse XF DMEM medium (pH 7.40), 25 mM glucose (Sigma-Aldrich D9434), and 2 mM glutamine (Sigma-Aldrich G8540). The cells were then incubated in 180 μl assay medium at 37˚C in a non-$CO_2$ incubator for 1 h.

A modified cell mitochondria stress assay was performed by sequential injection of 1 μm oligomycin (Abcam ab141829), 0.25 μm Carbonyl cyanide 4-(trifluoromethoxy) phenylhydrazone (FCCP, Sigma-Aldrich C2920), 1 μm rotenone (Sigma-Aldrich R8875) in combined with 1 μm antimycin A (Sigma-Aldrich A8674). OCR was automatically recorded and calculated using the Seahorse XFe96 extracellular flux analyzer.

## Statistical analysis

All experiments were repeated as indicated in each figure legend. All data in bar and line graphs are expressed as Means ± SDs. The statistical significance of differences between groups was determined using Student's unpaired *t* tests using Graphpad Prism 6 (Graphpad Software Inc.). A value of $p < 0.05$ was considered statistically significant (***$p < 0.001$; **$p < 0.01$; *$p < 0.05$; ns, not significant). *P* values were corrected to Q values using Benjamini and Hochberg correction in multiple tests.

## Supporting information

**S1 Table. Screen results of PS/PE pathway in *Drosophila* photoreceptor cells.**
(PDF)

**S2 Table. qPCR validation of indicate lines and S2 cell, Hela cells.**
(PDF)

**S1 Data. Underlying data for the main figures and the supplemental figures.**
(XLSX)

**S1 Raw Images. Raw images for all western blots.**
(PDF)

**S1 Fig. Effective suppression of *pect*^*RNAi5*^ by overexpression of PISD.** (A) *pect* or *pcyt1* deficiency failed to rescue *pss*^*RNAi*^ and *pisd*^*RNAi*^. Muscle sections from *EGFP*^*RNAi*^ (*MHC-gal4/UAS-EGFP*^*RNAi*^), *pss*^*RNAi*^ (*MHC-gal4/+;UAS-pss*^*RNAi*^/+), *pss*^*RNAi*^+*pect*^*RNAi*^ (*MHC-gal4/+;UAS-pss*^*RNAi*^/*UAS-pect*^*RNAi*^), *pss*^*RNAi*^+*pcyt1*^*RNAi*^ (*MHC-gal4/+;UAS-pss*^*RNAi*^/*UAS-pcyt1*^*RNAi*^), *pisd*^*RNAi*^ (*MHC-gal4/+;UAS-pisd*^*RNAi*^/+), *pisd*^*RNAi*^+*pect*^*RNAi*^ (*MHC-gal4/+;UAS-pisd*^*RNAi*^/*UAS-pect*^*RNAi*^), and *pisd*^*RNAi*^+*pcyt1*^*RNAi*^ (*MHC-gal4/+;UAS-pisd*^*RNAi*^/*UAS-pcyt1*^*RNAi*^) flies. Scale bar, 1 μm. Mitochondria size of at least 6 samples of each phenotype were quantified. (B) The ratio of mRNA level of *pect*, *pcyt1*, *pis* and *bbc* in *MHC>pect*^*RNAi*^, *MHC>pcyt1*^*RNAi*^, *MHC>pis*^*RNAi*^, and *MHC>bbc*^*RNAi*^ individually to *MHC>EGFP*^*RNAi*^. Total RNA was extracted from the dissected fly thorax of *MHC>EGFP*^*RNAi*^, *MHC>pect*^*RNAi*^, *MHC>pcyt1*^*RNAi*^, *MHC>pis*^*RNAi*^, and *MHC>bbc*^*RNAi*^. The relative expression of target genes was normalized to *RP49*, which serves as an internal control. Data are presented as mean ± SD, *$p < 0.05$ (Student's unpaired $t$ test). $n = 3$. (C) ERG recordings from 5-day-old control (*EGFP*^*RNAi*^, *GMR-gal4/UAS-EGFP*^*RNAi*^), *pect*^*RNAi1*^ (*GMR-gal4/UAS-pect*^*RNAi1*^), *pect*^*RNAi2*^ (*GMR-gal4/UAS-pect*^*RNAi2*^), *GMR>pect*^*RNAi2w*^ (*GMR-gal4/UAS-pect*^*RNAi2w*^), *GMR>pect*^*RNAi3*^ (*GMR-gal4/UAS-pect*^*RNAi3*^), *GMR>pect*^*RNAi5*^ (*GMR-gal4/UAS-pect*^*RNAi5*^), and *GMR>pect*^*RNAi6*^ (*GMR-gal4/UAS-pect*^*RNAi6*^) flies. Flies were exposed to a 5-s pulse of orange light after 2 min of dark adaptation. (D) mRNA level of *pect* was calculated of *pect*^*RNAi*^ in muscle and retina. Total RNA was extracted from the dissected fly thorax and retina of *MHC>EGFP*^*RNAi*^, *MHC>pect*^*RNAi5*^, *GMR>EGFP*^*RNAi*^, and *GMR>pect*^*RNAi5*^. The relative expression of target genes was normalized to *RP49*, which serves as an internal control. Data are presented as mean ± SD, *$p < 0.05$ (Student's unpaired $t$ test). $n = 3$. (E) Overexpression of PISD rescued the defects caused by *pect*^*RNAi*^ in ERG response. Five-day-old *EGFP*^*RNAi*^ (control), *pect*^*RNAi1*^+*pisd* (*trp-pisd GMR-gal4/UAS-pect*^*RNAi1*^), *pect*^*RNAi2*^+*pisd* (*trp-pisd GMR-gal4/UAS-pect*^*RNAi2*^), *pect*^*RNAi2w*^+*pisd* (*trp-pisd GMR-gal4/UAS-pect*^*RNAi2w*^), *pect*^*RNAi5*^+*pisd* (*trp-pisd GMR-gal4/UAS-pect*^*RNAi5*^), and *pect*^*RNAi6*^+*pisd* (*trp-pisd GMR-gal4/UAS-pect*^*RNAi6*^) files were exposed to a 5-s pulse of orange light. (F) Knocking down *serca* and *Marf* abolished the rescue effects of PISD towards *pect*. ERGs were recorded from 5-day-old *pect*^*RNAi5*^+*pisd*, *pect*^*RNAi5*^+*pisd*+*EGFP*^*RNAi*^ (*trp-pisd GMR-gal4/+;UAS-pect*^*RNAi5*^/UAS-EGFP*^*RNAi*^), *pect*^*RNAi5*^+*pisd*+*serca*^*RNAi*^ (*trp-pisd GMR-gal4/+;UAS-pect*^*RNAi5*^/UAS-serca*^*RNAi*^), *serca*^*RNAi*^ (*GMR-Gal4/UAS-serca*^*RNAi*^), *pect*^*RNAi5*^+*pisd*+*Marf*^*RNAi*^ (*trp-pisd GMR-gal4/+;UAS-pect*^*RNAi5*^/UAS-Marf*^*RNAi*^), and *Marf*^*RNAi*^ (*GMR-Gal4/UAS-Marf*^*RNAi*^). Flies were exposed to a 5-s pulse of orange light after 2 min of dark adaptation.
(TIF)

**S2 Fig. SLMO is required for the proper function of PISD.** (A) Schematic of the genome-wide RNAi screen for factors that prevent the rescue effects of PISD on the ERG defects of *pect*^*RNAi*^. (B) The *slmo* locus and mutation site are associated with *slmo*^*1*^. DNA sequencing revealed that *slmo*^*1*^ eliminates 28 bp within the *slmo* coding region. (C) *slmo*^*1*^ mutants are growth arrested. Images of *slmo*^*1*^ homozygous mutant larvae at different stages (24–96 h) of larval development. *w*^*1118*^ was used as a wild-type control. The scale bar is 100 μm. (D) Light microscope images of eyes of 1-day-old *w*^*1118*^ and *slmo*^*1*^ (*ey-flp;slmo*^*1*^ *FRT40A/GMR-hid CL FRT40A*) flies. The scale bar is 100 μm. (E) Strong *slmo*^*RNAi*^ lines induced severe functional and structural defects in photoreceptor cells. ERG recordings from 1-day-old control (*GMR-gal4/UAS-EGFP*^*RNAi*^), *slmo*^*RNAi1*^ (*GMR-gal4/UAS-slmo*^*RNAi1*^), *slmo*^*RNAi2*^ (*GMR-gal4/UAS-slmo*^*RNAi2*^), and *slmo*^*RNAi1*^+*slmo*^*R1*^ (*GMR-gal4/UAS-slmo*^*RNAi1*^ *UAS-slmo*^*R1*^/+) flies. Flies were exposed to a 5-s pulse of orange light after 2 min of dark adaptation. (F) The amplitudes and off-transients of ERG responses were quantified. At least 10 flies were used, and significant differences were determined using the unpaired $t$ test. (G) TEM retinal sections were obtained from 5-day-old flies of the same genotype as (E). The scale bar is 2 μm. (H) TEM sections of

muscles from *slmo^RNAi2^* (*MHC-gal4/UAS-slmo^RNAi2^*) and *slmo^RNAi^* (*MHC-gal4/UAS-slmo^RNAi^*) flies show decreased mitochondrial size and damaged cristae. The scale bar is 0.5 μm. The data underlying the graphs shown in the figure can be found in S2 Table.
(TIF)

**S3 Fig. The role of SLMO in the PSS/PISD pathway.** (A) Lipidomic analysis of mitochondrial PC, PE, and PS levels of control (*EGFP^RNAi^*, *MHC-gal4/UAS-GFP^RNAi^*), *pss^RNAi^* (*MHC-gal4/ UAS-pss^RNAi^*), *pss^RNAi^+slmo* (*MHC-gal4/UAS-pss^RNA^;UAS-slmo*), *slmo*(*MHC>UAS-slmo*), *pss^RNAi^+pss* (*MHC-gal4/UAS-pss^RNAi^;UAS-PSS*), and *pss* (*MHC-gal4/UAS-pss*) muscles. Mitochondria were isolated from 10 dissected thoraxes per assay, and 3 replicates were quantified. (B) Overexpression of *pisd* restored mitochondrial PE levels the *slmo^RNAi^* retina. Lipidomic analysis of mitochondrial PC, PE, and PS levels of *EGFP^RNAi^* (*GMR-gal4/UAS-GFP^RNAi^*), *slmo^RNAi^* (*GMR-gal4/UAS-slmo^RNAi^*), *slmo^RNAi^+trp-pisd* (*GMR-gal4/UAS-slmo^RNA^;trp-pisd*), and *trp-pisd* photoreceptor cells. Mitochondria were isolated from 60 dissected thoraxes per assay, and 3 replicates were quantified. (C) SLMO overexpression failed to reverse decreased mitochondrial size from *pisd^RNAi^* lines. TEM sections of muscles from *EGFP^RNAi^*, *pisd^RNAi^* (*MHC-gal4/UAS-pisd^RNAi^*), *slmo* (*MHC-gal4/UAS-slmo*), and *slmo+pisd^RNAi^* (*MHC-gal4/UAS-slmo;UAS-pisd^RNAi^*) flies. The scale bar is 2 μm. Mitochondria size was quantified and significant differences were determined using the unpaired *t* test.
(TIF)

**S4 Fig. Verification of split-GFP and immunostaining system.** (A) ERG recordings from 5-day-old *slmo^RNAi1^* (*GMR-gal4/UAS-slmo^RNAi1^*), *slmo^RNAi1^+ubi-slmo^R1^-GFP* (*GMR-gal4/+; UAS-slmo^RNAi1^/ubi-slmo^R1^-GFP*), and *ubi-slmo^R1^-GFP* (*ubi-slmo^R1^-GFP*) flies. (B) Light microscope images of eyes of 1-day-old *slmo^1^* (*ey-flp;slmo^1^ FRT40A/GMR-hid CL FRT40A*) and *slmo^1^+ ubi-slmo^R1^* (*ey-flp;slmo^1^ FRT40A/GMR-hid CL FRT40A;ubi-slmo^R1^-GFP/+*) flies. The scale bar is 100 μm. (C) S2 cells expressing SLMO-GFP were imaged using confocal fluorescent microscopy. TOM20 (red) and CNX99 (blue) were stained to visualize mitochondria and ER, respectively. Scale bar, 10 μm. (D) Cell lysates from S2 cells transfected with SDHC(1–94)-mCherry and mito-GFP (Cox8-Sod2-GFP, N-terminal sequences form Cox8 and SOD2 drives GFP to mitochondrial matrix) were analyzed using the protease K protection assay. Without Triton X-100, both mCherry and GFP were resistant to protease K treatment. (E) Design of the split-GFP system. spGFP1-10 was fused to the protein of interest (POI) and spGFP11 was fused to TOM20, SDHC(1–94), or SDHC(1–139). When the POI and spGFP11 were on the same side of the membrane, spGFP1-10 and spGFP11 bind and emit green fluorescence. (F) Verification of split-GFP system in S2 cells. S2 cells co-expressing sp1-10-tagged IMS protein MICU1 (MICU1-sp1-10) and matrix protein Cox8 with SDHC(1–94)-mCherry-sp11 or SDHC(1–139)-mCherry-sp11, were directly imaged for mCherry (red) and GFP (green) fluorescence using confocal fluorescent microscopy. The sp1-10 tag was stained with rabbit-GFP antibody (blue) as well. Scale bar, 10 μm. (G) Schematic diagram of the *slmo-HA* knock-in strategy. *slmo-HA* knock-in flies were verified via genomic DNA sequencing. (H) Muscle tissues from *ubi-SDHC-GFP* and *ubi-TOM20-GFP* were penetrated by 5 μg/ml or 0.1 mg/ml digitonin, and the samples were stained for TOM20-GFP/SDHC-GFP (red), and GFP signals were directly observed (green). Scale bar, 10 μm.
(TIF)

**S5 Fig. Transport of PS from the OMM to the IMM by SLMO is required to maintain mitochondrial morphology and function.** (A) Purification of MBP-SLMO and MBP from *E. coli* cell lysates. The supernatant, pellet, flow through, and elution were analyzed by SDS–PAGE. Asterisks (*) mark MBP-SLMO and MBP. (B) A schematic diagram of the lipid membrane

strip containing the indicated phospholipids is shown on the left. Grip-GST, which specifically binds PI (4, 5) P, was used as a positive control. MBP-SLMO specifically interacts with PS, and MBP alone did not interact with any phospholipid. (C) Liposome flotation assay shows that SLMO binds PS. The supernatants and pellets (100 µl of each sample, from top to down) were subjected to Tricine-SDS PAGE analysis. (D) Amino acid alignment of yeast UPS2, SLMO (*Drosophila*), and SLMO2 (Human) using the ClustalW Multiple Sequence Alignment program. Identical and conserved residues are indicated with asterisks and dots, respectively. T93A and N150A are circled. (E) Liposome flotation assay of SLMO and 2 mutants. T93A and N150A both reduced the binding affinity between SLMO and PS. (F) Muscle tissues from *DA-slmo$^{N150A}$-GFP* or *DA-slmo$^{T93A}$-GFP* were dissected and stained with TOM20 (red) and GFP (green). Scale bar, 10 µm. (G) Western blot analysis of proteins extracted from the heads of *w$^{1118}$*, *DA-slmo-GFP*, *DA-slmo$^{N150A}$-GFP*, and *DA-slmo$^{T93A}$-GFP* flies with antibodies against GFP. Actin was used as a loading control. (H) Light microscope images of eyes of 1-day-old *slmo$^1$* (*ey-flp;slmo$^1$ FRT40A/GMR-hid CL FRT40A*), *slmo$^1$+slmo$^{N150A}$* (*ey-flp;slmo$^1$ FRT40A/ GMR-hid CL FRT40A;DA-slmo$^{N150A}$/+*), *slmo$^1$+slmo$^{T93A}$* (*ey-flp;slmo$^1$ FRT40A/GMR-hid CL FRT40A;DA-slmo$^{T93A}$/+*), and *slmo$^1$+slmo* (*ey-flp;slmo$^1$ FRT40A/GMR-hid CL FRT40A;DA-slmo/+*) flies. Scale bar, 100 µm. (I) ERG recordings from 5-day-old control (*GMR-gal4/ EGFP$^{RNAi}$*), *slmo$^{RNAi1}$* (*GMR-gal4/UAS-slmo$^{RNAi1}$*), *slmo$^{RNAi1}$+EGFP* (*GMR-gal4/+;UAS-slmo$^{RNAi1}$/UAS-EGFP*), *slmo$^{RNAi1}$+slmo$^{N150A}$* (*GMR-gal4/+;UAS-slmo$^{RNAi1}$/DA-slmo$^{N150A}$*), *slmo$^{RNAi1}$+slmo$^{T93A}$* (*GMR-gal4/+;UAS-slmo$^{RNAi1}$/DA-slmo$^{T93A}$*) flies.
(TIF)

**S6 Fig. SLMO can transfer PS in vivo and is independent of mitochondrial dynamics.** (A) Images of cells transfected with GFP-LactC1C2 (green) with TOM20-PSS-RFP (red) or dsRNA of *pss* (*pss$^{RNAi}$*) or *pisd* (*pisd$^{RNAi}$*). Scale bar, 10 µm. (B) Confocal (up) and Super-resolution SIM microscopy (down) show the localization of LactC1C2, TOM20, and Cox8. S2 cells were transfected with dsRNA of *slmo$^{RNAi}$*, pIB-GFP-LactC1C2, and pIB-Cox8-HA, stained by antibody against TOM20 and HA tag. Scale bar, 10 µm. (C) TEM sections of indirect flight muscles of control (*MHC-gal4/UAS-GFP$^{RNAi}$*), *slmo$^{RNAi2}$+Drp1* (*MHC-gal4/+;UAS-Drp1/ UAS-slmo$^{RNAi2}$*), *Drp1* (*MHC-gal4/UAS-drp1*), *slmo$^{RNAi2}$* (*MHC-gal4/UAS-slmo$^{RNAi2}$*), and *slmo$^{RNAi2}$+Mfn* (*MHC-gal4/+;UAS-slmo$^{RNAi2}$/UAS-marf*) and *Mfn* (*MHC-gal4;UAS-marf*) flies. Scale bar, 2 µm. All flies were raised for 5 days under 12 h-light/12 h-dark cycles. (D) Muscle section of control (*MHC-gal4/UAS-GFP$^{RNAi}$*), *slmo$^{RNAi1}$* (*MHC-gal4/+;UAS-slmo$^{RNAi1}$/ +*), *opa1* (*MHC-gal4/+;UAS-opa1/+*), and *slmo$^{RNAi1}$+opa1* (*MHC-gal4/+;UAS-slmo$^{RNAi2}$/UAS-opa1*) flies. Scale bar, 1 µm.
(TIF)

**S7 Fig. The function of SLMO is independent of dTRIAP1/2.** (A) Overexpression of *dtriap2* did not affect *slmo$^{RNAi}$*. ERG recordings from 5-day-old control (*GMR-gal4/UAS-EGFP$^{RNAi}$*), *dtriap2* (*GMR-gal4/UAS-dtriap2*), *slmo$^{RNAi1}$* (*GMR-gal4/UAS-slmo$^{RNAi1}$*), and *slmo$^{RNAi1}$+d-triap2* (*GMR-gal4/+;UAS-slmo$^{RNAi1}$/UAS-dtriap2*) flies. (B) Co-expression of dTRIAP1/2 and SLMO did not affect mitochondria size. TEM sections from control (*MHC-gal4/UAS-RFP*), *dtriap2* (*MHC-gal4/UAS-dtriap2*), *dtriap2+slmo* (*MHC-gal4/+;UAS-slmo/UAS-dtriap2*), *slmo* (*MHC-gal4/UAS-slmo*), *dtriap1* (*MHC-gal4/UAS-dtriap1*), and *dtriap1+slmo* (*MHC-gal4/+; UAS-slmo/UAS-dtriap1*) flies. (C) The RNAi efficiency was determined using quantitative Real-Time PCR (qPCR). Total RNA was extracted from the retina dissected from *EGFP$^{RNAi}$* (*GMR-gal4/+;EGFP$^{RNAi}$/+*) or *dtriap1/2$^{RNAi}$* (*GMR-gal4/+;UAS-dtriap1/2$^{RNAi}$*) flies. The relative expression of target genes was normalized to *RP49*, which served as an internal control. Data are presented as mean ± SD, *$p < 0.05$, ***$p < 0.005$ (Student's unpaired *t* test). $n = 3$. (D) *dTRIAP1/2$^{RNAi}$* failed to reduce the expression of SLMO-HA. SLMO-HA protein levels

are indicated using western blotting of 2 thoraxes from 5-day-old $w^{1118}$, *slmo-HA(slmo-HA/+)*, or s*lmo-HA+MHC>dTRIAP1/2$^{RNAi}$* (*MHC-gal4/slmo-HA;UAS-dTRIAP1/2$^{RNAi}$*) flies. (E) Muscle sections from control (*MHC-gal4/UAS-GFP$^{RNAi}$*), *dtriap1/2$^{RNAi}$* (*MHC-gal4/+;UAS-dtriap1/2$^{RNAi}$/+*), *slmo$^{RNAi2}$* (*MHC-gal4/UAS-slmo$^{RNAi2}$*), and *slmo$^{RNAi2}$+dtriap1/2$^{RNAi}$* (*MHC-gal4/UAS-slmo$^{RNAi2}$;UAS-dtriap1/2$^{RNAi}$/+*) flies. The data underlying the graphs shown in the figure can be found in S2 Table.
(TIF)

**S8 Fig. Mammalian SLOM2 can perform SLMO-conserved functions in flies.** (A) Quantification of the amplitude of ERG responses and off transients from 5-day-old control (*GMR-Gal4/UAS-EGFP$^{RNAi}$*), *slmo$^{RNAi1}$* (*GMR-Gal4/UAS-slmo$^{RNAi1}$*), *slmo$^{RNAi1}$+EGFP* (*GMR-Gal4/+;UAS-slmo$^{RNAi1}$/UAS-EGFP*), *slmo$^{RNAi}$+PRELID1* (*GMR-Gal4/+;UAS-slmo$^{RNAi1}$/UAS-PRELID1*), *slmo$^{RNAi1}$+SLMO1* (*GMR-Gal4/+;UAS-slmo$^{RNAi1}$/UAS-SLMO1*), and *slmo$^{RNAi1}$+SLMO2* (*GMR-Gal4/+;UAS-slmo$^{RNAi1}$/UAS-SLMO2*) flies. At least 6 flies of each genotype were analyzed. (B) SLMO1 does not localize to mitochondria. Hela cells were transfected with SLMO1-GFP and stained with TOM20 (red) antibodies. Scale bar, 10 μm. (C) Verification of the split-GFP system in Hela cell. Hela cells co-expressing sp1-10-tagged IMS protein MICU1 (MICU1-sp1-10) with SDHC(1–94)-mCherry-sp11 or SDHC(1–139)-mCherry-sp11 were directly imaged for GFP (green) and mCherry fluorescence (red). Scale bar, 10 μm. (D, E) Oxygen consumption rates (OCR) in Hela cells transfected with *Slmo2$^{RNAi1}$*, *Slmo2$^{RNAi2}$*, *Slmo2$^{RNAi1}$+SLMO2*, and *Slmo2$^{RNAi2}$+SLMO2*. Representative kinetics graph indicating real-time OCR at baseline and after addition of oligomycin, carbonyl cyanide p-tri-fluoromethoxy-phenylhydrazone (FCCP), and rotenone-antimycin (R/A). Basal respiration and proton leak of the indicated genotypes were calculated by normalization of OCR levels to ATP levels. $n$ = 3 wells/group. (F) Quantification of mitochondrial fusion and fission rates in Hela cells. Cells expressing Cox8-GFP transfected with *Slmo2$^{RNAi1}$* and *Mfn1* knock-out cells were used as controls. The data underlying the graphs shown in the figure can be found in S2 Table.
(TIF)

## Acknowledgments

We thank the Bloomington Stock Center, the *Drosophila* Genomic Resource Center, the Tsinghua Fly Center, and the Developmental Studies Hybridoma Bank, for stocks and reagents. We thank Y. Wang and X. Liu for assistance with fly injections, Dr. Yan Ma from the Metabolomics Center for lipidomic mass spectrometric analysis, Xiumei Jin from the Electron Microscopy Center for sample preparation of TEM, the Sequencing Center, the Imaging Facility, and the Proteomics Facility at the Institute of Biological Sciences for technique support. We thank Dr. Lin Yang from the Institute of Genetics and Developmental Biology for assistance with TEM. We thank Dr. D. O'Keefe for editing the manuscript.

## Author Contributions

**Conceptualization:** Siwen Zhao, Ning Li, Tao Wang.

**Investigation:** Siwen Zhao.

**Methodology:** Siwen Zhao, Xuguang Jiang, Ning Li.

**Supervision:** Tao Wang.

**Visualization:** Xuguang Jiang.

**Writing – original draft:** Siwen Zhao, Tao Wang.

**Writing – review & editing:** Siwen Zhao, Tao Wang.

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
