## [Editor Report · Decision Letter 0]

25 Apr 2024

Dear Dr Wang, 

Thank you for submitting your manuscript entitled "Phosphatidylserine transfer by PSS/SLMO/PISD is critical for mitochondrial organization and function" for consideration as a Research Article by PLOS Biology.

Your manuscript has now been evaluated by the PLOS Biology editorial staff as well as by an academic editor with relevant expertise and I am writing to let you know that we would like to send your submission out for external peer review.

Once your full submission is complete, your paper will undergo a series of checks in preparation for peer review. After your manuscript has passed the checks it will be sent out for review. To provide the metadata for your submission, please Login to Editorial Manager (https://www.editorialmanager.com/pbiology) within two working days, i.e. by Apr 29 2024 11:59PM.

Kind regards,

Ines

--

Ines Alvarez-Garcia, PhD

Senior Editor

PLOS Biology

---

## [Decision Letter · Decision Letter 1]

27 Jun 2024

Dear Dr Wang,

Thank you for your patience while your manuscript entitled "Phosphatidylserine transfer by PSS/SLMO/PISD is critical for mitochondrial organization and function" was peer-reviewed at PLOS Biology. Please also accept my apologies for the delay in sending you our decision. The manuscript has been evaluated by the PLOS Biology editors, an Academic Editor with relevant expertise, and by three independent reviewers. 

The reviews are attached below. As you will see, the reviewers find the conclusions interesting, however they also raise several issues that would need to be addressed before we can consider the manuscript for publication. Reviewer 1 thinks that some of the claims should toned down acknowledging the work done on SLMO in mammals, and that quantifications of every reiteration of knockdown and re-expression experiments should be performed. Reviewer 2 asks for additional controls to support the conclusion that SLMO occupies the inner boundary space, and also raises several points that should be clarified. Reviewer 3 would like to see confirmation of the epistatic relationship between SLMO and PSS/PISD and also data supporting the localisation of SLMO in IBM, as it seems to be missing.

In light of the reviews, which you will find at the end of this email, we would like to invite you to revise the work to thoroughly address the reviewers' reports. Given the extent of revision needed, we cannot make a decision about publication until we have seen the revised manuscript and your response to the reviewers' comments. Your revised manuscript is likely to be sent for further evaluation by all or a subset of the reviewers.

**IMPORTANT - SUBMITTING YOUR REVISION**

3. Resubmission Checklist

a) *PLOS Data Policy*

b) *Published Peer Review*

d) *Blurb*

Please also provide a blurb which (if accepted) will be included in our weekly and monthly Electronic Table of Contents, sent out to readers of PLOS Biology, and may be used to promote your article in social media. The blurb should be about 30-40 words long and is subject to editorial changes. It should, without exaggeration, entice people to read your manuscript. It should not be redundant with the title and should not contain acronyms or abbreviations. For examples, view our author guidelines: https://journals.plos.org/plosbiology/s/revising-your-manuscript#loc-blurb

Sincerely,

Ines

--

Ines Alvarez-Garcia, PhD

Senior Editor

PLOS Biology

Reviewers' comments

Rev. 1:

Zhao et al. studied phospholipid trafficking between the ER and mitochondria using drosophila melanogaster. They focused on mitochondrial PS transport, and using a forward genetic screen identified Slowmo (SLMO) as a PS transporter important for mitochondrial biogenesis. A major criticism with this narrative, is that mammalian isoforms of SLMO (Slmo1 and Slmo2) have already been previously identified as mitochondrial phospholipid transporter in mammals, yet the manuscript barely recognizes this. It is not to say that the paper lacks in novelty: in particular, Figure 5 and 6 shows data on SLMO binding to PS, and the potential non-essential role of TRIAP in drosophila. In general, experiments are well designed, executed, and data presented well, but the whole tone of the manuscript needs to be changed. Authors need to better argue how their findings in drosophila complement what is already known about Slmo2 (Prelid3b) in mammals.

Major comments:

* The authors overstate the novelty of the study findings as they relate to the existing knowledge of PS mitochondrial import. It is recommended that the authors make substantial revision on the narrative to acknowledge work done by Thomas Langer's group that shows Prelid3b (Slmo2) in mitochondrial PS import.

* Similarly, while cited, Jean Vance's group have substantial data on the role of PSS/PISD to show this pathway is essential for mitochondria. This is largely understated in the manuscript.

* The main deficient in data for this manuscript are the mitochondrial lipidomic data shown in Figure 1D, but not for any of the other subsequent figures. These quantifications need to be done on every subsequent reiterations of knockdown and reexpression experiments. Without this, authors cannot attribute changes in function or mitochondrial morphology that happens with changes in PSS/SLMO/PISD or TRIAP axes.

* Figure 3: how is PISD overexpression rescuing defects induced by Slmo knockdown? If Pisd is downstream of Slmo, PISD overexpression might presumably not rescue any of these phenotype (as there is no mitochondrial PS available as a substrate for PISD). This is one of the reasons why lipidomic data are needed.

* Figure 5G and S4H: Please also show the colocalization of PS with IMM marker, not just OMM (TOM20). I am not really buying the argument that SLMO deletion promotes accumulation of PSS on OMM according to these images alone.

Minor comments:

* Line 228 please change S5B to S4B

* Line1213 No schematic is found as indicated in the figure caption

Rev. 2:

Mitochondria are essential organelles that have two membranes that differ in their protein and lipid compositions. The mitochondrial inner membrane contains enzymes that result in the production of cardiolipin and phosphatidylethanolamine (PE). Both lipid biosynthetic pathways are fed by precursor phospholipids that originate mainly in the endoplasmic reticulum. In addition, the ER is the source for the other major phospholipid classes that populate mitochondrial membranes but are not made in this organelle. Mechanisms by which mitochondria obtain phospholipids from the ER and move specific phospholipids between the outer and inner membranes have begun to emerge over the past ~15 years. In the present study, Zhao et al provide compelling genetic evidence that the proper synthesis of ER-generated phosphatidylserine (PS) by PSS and its subsequent conversion to PE by PISD in the inner membrane is important for mitochondrial structure and function in Drosophila. To identify factors with roles in connecting PSS and PISD activities, a genome wide RNAi screen was performed which identified SLMO, a fly ortholog of the Ups/Preli family which has well documented roles in PS and phosphatidic acid transport from the OM to the IM. Through epistasis analyses, SLMO was placed downstream of PSS and upstream of PISD, and consistent with this placement, shown to reside in the intermembrane space. In yeast and mammals, Ups/Preli proteins heterodimerize with Mdm35/TRIAP1 and this interaction is both important for their lipid transport functions and their stability. In contrast to this expectation, fly SLMO can transport PS between liposomes alone in vitro and genomically HA tagged SLMO steady state amounts and localization to mitochondria are unaffected by the absence of both predicted fly TRIAP proteins. Surprisingly, while both dTRIAP proteins localize to mitochondria, they are associated with the cytosol-facing surface of the OM. Expression of human SLMO2, but not PRELID1 or SLMO1, rescued slmoRNAi1 fly phenotypes, suggesting that SLMO2 can also transport PS from the OM to the IM. Hela cells depleted of SLMO2 had smaller mitochondria with aberrant cristae, similar to what was observed in slmo-knockdown flies. Overall, this is an elegant and well controlled study that provides compelling evidence implicating SLMO in the movement of PS into the IM across the intermembrane space. The main twist presented relates to the apparent ability of SLMO to do this solo, an observation with regulatory implications.

Main points:

1. Additional controls are needed to support the conclusion that SLMO occupies the inner boundary space. Do import components also give the same staining pattern as SLMO-HA? Figure 4D would benefit from better labelling (e.g which panels represent TOM20 staining).

2. Lines 219-221: "Using super-resolution microscopy, we found a circular pattern of SLMO-HA that co-localized with TOM20. By contrast, the IMM protein SDHC, which is enriched in the CM region, was surrounded by TOM20 (Fig 4D)." It is unclear whether this data is shown in Figure 4D given that TOM20 panels aren't clearly labeled. If this data is present, it does not support the conclusion that SDHC staining is surrounded by TOM20.

3. Are the 2 SLMO mutants deficient in PS binding as predicted?

4. Given that the Ups/Preli family from yeast and mammals are short-lived due to their continuous degradation by Yme1, this would suggest that this pathway is regulated in a distinct manner in flies. Given this difference, it might be worth adding this possibility to the discussion section.

5. Is the PE percentage in mitochondria really less than 1% of total phospholipids? And the amount of PC is less than PE? The amount of each seems very low and inconsistent with the contention that PE and PC are major building blocks of membranes. If PE and PC make up less than 2% of mitochondrial lipids, what are the main components of its membranes? Some explanation for these values would be helpful.

Minor points:

1. The GFP signal in Fig. 7D is pretty weak compared to the other split-GFP-based data. Perhaps a more convincing image could be provided.

2. Line 228: (Fig. S5B) should be changed to (Fig. S4B).

3. Line 229: This assay seems like a liposome floatation assay, not a liposome sediment assay. Also, I could not find the method for this assay (Fig. s4c).

4. Abstract: Would add "putative" to sentence: "SLMO is required for shaping mitochondrial morphology, but its putative conserved binding partner, dTRIAP, is not." This better reflects the authors'discovery that both dTriap proteins are on the OM and that SLMO protein levels are insensitive to their combined absence.

5. Lines 45-47: This statement is incorrect as mitochondria are the sole organelle that makes cardiolipin. Suggest changing "The only" to "An exception".

6. Lines 64-66: This statement is also incorrect as several papers have assessed mitochondrial morphology and function in the context of perturbed phospholipid movement into and across the mitochondrial intermembrane space, including several studies that the authors' have cited (PMID: 23042293; PMID: 27241913; PMID: 29301859; PMID: 23931759; PMID: 30650346).

7. For the non-fly expert, it would be helpful if the EM images were labeled more thoroughly (e.g. the rhabodomeres and photoreceptor cells; mitochondria and sarcomeres?).

8. What side of the IM is the GFP tag attached to SDHC?

Rev. 3:

Zhao et al. utilized a forward genetic screen in Drosophila to explore how PS-PE production in mitochondria impacts mitochondrial function and structure. They identified the protein Slowmo (SLMO) from a sensitized background. Through microscopy, ERG, TEM, and lipidomic analysis, SLMO was found to be crucial for PSS/PISD in maintaining mitochondrial structure and function. They demonstrated that SLMO specifically transfers PS from the outer mitochondrial membrane to the inner mitochondrial membrane via an in vitro lipid transfer/binding assay. This transport is vital for mitochondrial morphology, independent of mitochondrial dynamics. The authors further showed that the mammalian homologue SLMO2 is functionally conserved. The manuscript is well-written, with clearly presented data and well-controlled experiments. Combined with the smooth logical flow, this makes the manuscript an enjoyable read. This will be of interest to those working in related fields.

Some minor points:

- Why is PC increased in 1D? Can the authors explain the potential cause?

- The paragraph describing the epistatic relationship between SLMO and PSS/PISD (line 161 onward) seems weaker compared to other data. For example, the fact that slmoRNAi+PISD rescues the slmoRNAi phenotype does not eliminate the possibility that they function in parallel pathways rather than in an upstream/downstream relationship. I wonder whether the authors could show SLMO(OE)+PISD RNAi to further support their conclusion (the same applies to SLMO and PSS). Alternatively, the authors could modify the manuscript to acknowledge this possibility.

- The authors claimed that SLMO-GFP does not colocalize with TOM20 in mature sperm after elongation, due to the lack of a proper mitochondrial marker in the mature sperm tail. This description is somewhat tautological. Perhaps rephrase this explanation? (If the authors choose to present this, please add an arrow in Figure 4A to indicate the mature sperm.)

- The data supporting the localization of SLMO in IBM seem to be missing (line 219 and Fig 4D). Please elaborate on the logic or data that support the idea (especially focus on how they can distinguish cristae vs IBM).

- Are the SLMO T93 and N150 residues conserved? Can the authors include some alignments? Are there known diseases associated with mutations in these residues?

- The PS binding assay uses LactC1C2 as a reporter. Can the authors provide more information about which part of PS it binds/recognizes (e.g., head group)? If I understood correctly, the LactC1C2 was expressed without a mitochondrial targeting sequence, limiting the reporter to detect only the cytoplasmic side of the lipid bilayer of the OMM. The authors should make such information more explicit in writing.

- (Line 267) How did the authors conclude that PS trafficking to the plasma membrane is not affected by SLMO-KD? To interpret this alongside Figure S4H, why does PSS KD cause increased PS on the OMM and PISD KD cause increased PS in the plasma membrane? If this conclusion is based on confocal images, please include arrows and/or quantifications.

---

## [Decision Letter · Decision Letter 2]

2 Nov 2024

Dear Dr Wang,

Thank you for your patience while we considered your revised manuscript entitled "Phosphatidylserine transfer by PSS/SLMO/PISD is critical for mitochondrial organization and function" for publication as a Research Article at PLOS Biology. This revised version of your manuscript has been evaluated by the PLOS Biology editors, the Academic Editor and two of the original reviewers.

Based on the reviews, we are likely to accept this manuscript for publication, provided you satisfactorily address the data and other policy-related requests stated below.

In addition, we would like you to consider a suggestion to improve the title:

"SLMO transfers phosphatidylserine between the outer and inner mitochondrial membrane in Drosophila"

We expect to receive your revised manuscript within two weeks. 

*Published Peer Review History*

*Press*

Sincerely,

Ines

--

Ines Alvarez-Garcia, PhD

Senior Editor

PLOS Biology

Fig. 1C-F; Fig. 2B, D, F, G; Fig. 3B, E, G; Fig. 5A, C, E-G; Fig. 6D, E, F; Fig. 7F, G; Fig. S1A, B, D; Fig. S2F; Fig. S3A-C; Fig. S4; Fig. S7C and Fig. S8A, D, E, F

CODE POLICY

We require the original, uncropped and minimally adjusted images supporting all blot and gel results reported in an article's figures or Supporting Information files. We will require these files before a manuscript can be accepted so please prepare and upload them now. Please carefully read our guidelines for how to prepare and upload this data: https://journals.plos.org/plosbiology/s/figures#loc-blot-and-gel-reporting-requirements

We require the raw files for the gels shown in the following figures:

Fig. 4B; Fig. 6I; Fig. S4D; Fig. S5A, C, E, G and Fig. S7E

Reviewers' comments

Rev. 1:

No further comments

Rev. 3:

The initially submitted manuscript was already of commendable quality. The revisions further addressed all of my concerns. I support its publication in PLoS Biology.

---

## [Editor Report · Decision Letter 3]

15 Nov 2024

Dear Dr Wang,

Thank you for the submission of your revised Research Article entitled "SLMO transfers phosphatidylserine between the outer and inner mitochondrial membrane in Drosophila" for publication in PLOS Biology. On behalf of my colleagues and the Academic Editor, Alex Gould, I am delighted to let you know that we can in principle accept your manuscript for publication, provided you address any remaining formatting and reporting issues. These will be detailed in an email you should receive within 2-3 business days from our colleagues in the journal operations team; no action is required from you until then. Please note that we will not be able to formally accept your manuscript and schedule it for publication until you have completed any requested changes.

PRESS

Sincerely, 

Ines

--

Ines Alvarez-Garcia, PhD

Senior Editor

PLOS Biology
